# In situ n-doped nanocrystalline electron-injection-layer for general-lighting quantum-dot LEDs

Yizhen Zheng[1,4], Xing Lin [2,4] ✉, Jiongzhao Li[1], Jianan Chen[1], Wenhao Wu[1], Zixuan Song[2], Yuan Gao[3], Zhuang Hu[1], Huifeng Wang[1], Zikang Ye [1], Haiyan Qin [1] & Xiaogang Peng [1] ✉

Quantum-dot optoelectronics, pivotal for lighting, lasing and photovoltaics, rely on nanocrystalline oxide electron-injection layer. Here, we discover that the prevalent surface magnesium-modified zinc oxide electron-injection layer possesses poor n-type attributes, leading to the suboptimal and encapsulation-resin-sensitive performance of quantum-dot light-emitting diodes. A heavily n-doped nanocrystalline electron-injection layer—exhibiting ohmic transport with 1000 times higher electron conductivity and improved hole blockage—is developed via a simple reductive treatment. The resulting sub-bandgap-driven quantum-dot light-emitting diodes exhibit optimal efficiency and extraordinarily-high brightness, surpassing current benchmarks by at least 2.6-fold, and reaching levels suitable for quantum-dot laser diodes with only modest bias. This breakthrough further empowers white-lighting quantum-dot light-emitting diodes to exceed the 2035 U.S. Department of Energy's targets for general lighting, which currently accounts for ~15% of global electricity consumption. Our work opens a door for understanding and optimizing carrier transport in nanocrystalline semiconductors shared by various types of solution-processed optoelectronic devices.

Nanocrystalline ZnO and surface magnesium-modified ZnO (ZnMgO) thin films dominate as electron injection layers (EILs) in a variety of solution-processed optoelectronic devices, including quantum dot (QD) light-emitting-diodes[1-4] (QLEDs), perovskite LEDs[5], QD laser diodes[6], QD photodiodes[7] and solar cells[8]. In the case of QLEDs, the n-type attributes of the nanocrystalline EIL are crucial not only for electron conduction/injection and hole blockage but also for facilitating the challenging hole injection[9]. However, conventional doping strategies for bulk semiconductors are not effective for nanocrystalline EILs due to the high formation energy of dopants within the nanocrystalline lattice[10], and the properties are poorly understood and difficult to control. For instance, state-of-the-art QLEDs require encapsulation with

a resin containing volatile acids[11,12], which leads to erratic positive aging. This phenomenon is presumably caused by the prolonged and uncontrollable tuning of the EIL's n-type characteristics.

General lighting, which currently relies on electric and optical down-conversion emission with low power efficiency, accounts for >15% global electricity consumption[13]. The energy efficiency of an LED is best captured by its external power efficiency (EPE) with three components: the light-extraction efficiency (LEE), the internal quantum efficiency (IQE), and the ratio of the bandgap voltage $V_p$ ($V_p = h\nu/e$, with $h\nu$ as the average emitting photon energy and $e$ as elementary charge) to the driving voltage $V$, expressed as $EPE = LEE \times IQE \times V_p/V = EQE \times V_p/V$ (EQE, external quantum efficiency). Achieving high EPE is primarily

[1]Zhejiang Key Laboratory of Excited-State Energy Conversion and Energy Storage, and Department of Chemistry, Zhejiang University, Hangzhou, China.
[2]Zhejiang Key Laboratory of Excited-State Energy Conversion and Energy Storage, and College of Information Science and Electronic Engineering, Zhejiang University, Hangzhou, China. [3]Najing Technology Corporation Ltd., Hangzhou, China. [4]These authors contributed equally: Yizhen Zheng, Xing Lin. ✉e-mail: lxing@zju.edu.cn; xpeng@zju.edu.cn

about maximizing luminance at the lowest possible driving voltage while maintaining high IQE. GaN quantum well LEDs with phosphors and OLEDs suffer from low IQE and/or suboptimal $V_p/V$ ratios at the necessary luminance levels, resulting in a theoretical EPE of about 30%. QLEDs operating with sub-bandgap voltages offers a potential solution to minimize the down-conversion energy loss[14,15] in general lighting. Despite a decade of progress[2–4,14–26], fabrication of energy-efficient QLEDs with lighting-grade brightness remains elusive.

Our analysis suggests that the suboptimal n-type characteristics of the nanocrystalline EIL are the roadblock for QLEDs to become a main player in general lighting. Here, we demonstrate stable, heavily n-doped nanocrystalline EILs that achieve ideal ohmic electron transport, with a conductivity that is 3-6 orders of magnitude greater than that of the untreated one, addressing the long-standing issue of uncontrollable positive aging induced by acid-containing encapsulation. This is achieved through a simple and robust in situ reductive doping strategy for the nanocrystalline EIL, involving the state-filling of free electrons into the quantum-confined electron states of ZnMgO (or other oxide) nanocrystals. By boosting electron conductivity and hole blockage of the EIL, the resulting QLEDs across the entire visible spectrum simultaneously meet an optimal level for all three key parameters (IQE, luminescence, and $V_p/V$) required for general lighting, leading to a remarkable brightness and power efficiency. The developed EIL paves the way for sub-bandgap-voltage-driven QD-blend LEDs for diffuse white-light sources, surpassing the 2035 targets set by the U.S. Department of Energy for energy efficiency and performance attributes. Our findings offer critical insights for doping nanocrystalline semiconductor layers, a universal challenge among various solution-processed optoelectronic devices.

## Results

### The n-type doping of magnesium-modified zinc oxide thin film

The indispensable nanocrystalline oxide EILs in QD optoelectronics are assumed to be n-doped semiconductors as their bulk counterparts.

However, to our knowledge, their n-type properties in thin film form have rarely been investigated. After ~6 h of ultraviolet irradiation, a distinct electron-paramagnetic-resonance (EPR) signal, with a g-factor of 1.96–1.98, emerges in pristine ZnMgO thin films (Fig. 1a). This signal is attributed to the state-filling of free electrons in the quantum-confined band-edge electron states in ZnMgO nanocrystals[27,28]. Without ultraviolet irradiation, no discernible EPR signal is detected in pristine ZnMgO thin films regardless of water-vapor treatment. These findings hold true for both solution (Supplementary Fig. 1a) and film (Fig. 1a) forms, and are accompanied by the bleaching of the band-edge ultraviolet-visible absorption and the emergence of intra-band infrared absorption[27,29] (Supplementary Fig. 1b). These observations confirm the in situ photochemical n-doping of ZnMgO nanocrystals, revealing that pristine ZnMgO behaves as nearly an intrinsic semiconductor.

Photochemical doping relies on an inefficient hole transfer from the top of valance band of a photo-excited ZnMgO nanocrystal to a hole acceptor (such as ethanol or water)[27], which necessitates prolonged (several hours) ultraviolet irradiation. This requirement is impractical for large-scale production and detrimental to other functional layers in devices. To exclude intensive ultraviolet irradiation, in situ chemical n-doping—donating electrons directly from highly-reducing species into the quantized electron states at the bottom of conduction band of the nanocrystals in thin films—should be developed.

The deposition of a highly-reducing Al electrode onto a ZnMgO thin film (ZnMgO/Al film) yields a weak EPR signal with a g-factor of ~1.97 (Fig. 1b), suggesting a small number of free electrons have likely been transferred to the quantized electron levels of the top ZnMgO layer near the ZnMgO-Al interface. A subsequent water-vapor treatment for ~0.5 h leads to a huge characteristic EPR signal of free electrons in the quantized electron states of the nanocrystals in the ZnMgO/Al film. Our experimental and analytical results suggest that the observed n-type doping efficiency stems from reducing hydrogen

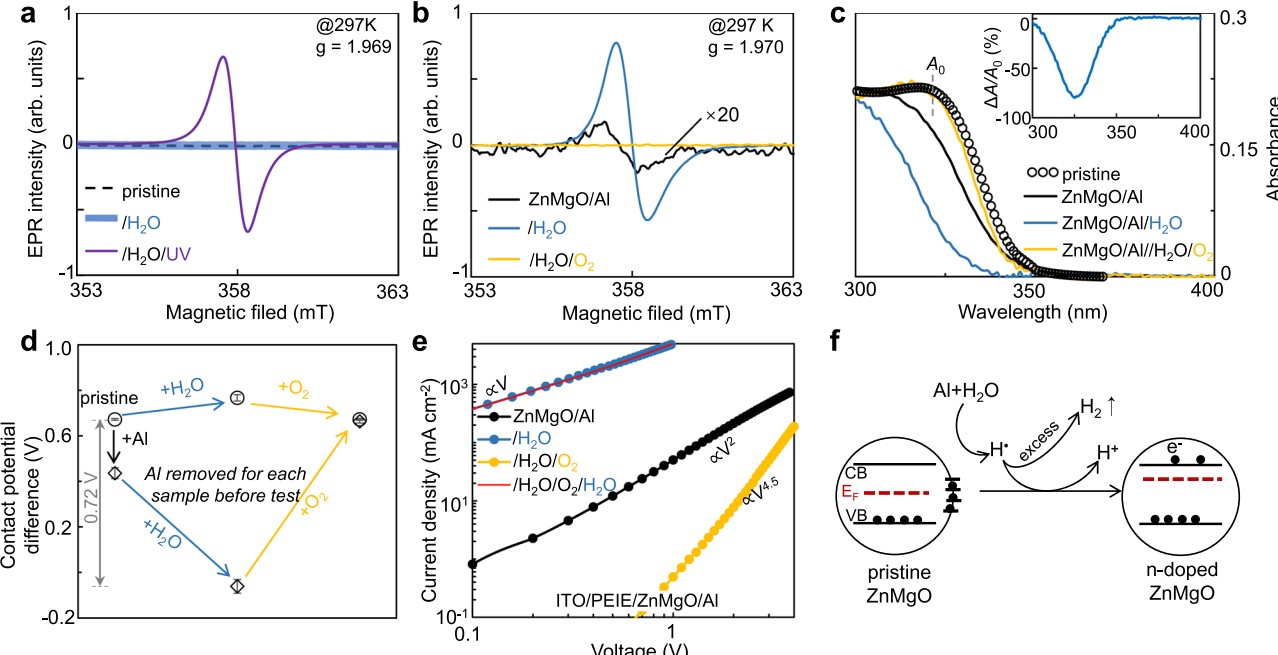

**Fig. 1 | In situ n-type doping in ZnMgO layer. a** EPR spectra of pristine ZnMgO film, after water-vapor treatment and after ultraviolet irradiation. **b** EPR spectra of ZnMgO/Al film after different treatments. **c** Absorption spectra of ZnMgO/Al film and the ones after water-vapor and oxygen treatments. Absorption spectrum of pristine ZnMgO in ethanol solution (black circle) is shown for reference. Inset: Difference absorption spectrum shows ~80% bleaching of band-edge absorption peak after water-vapor treatment. **d** Surface potential of ZnMgO and ZnMgO/Al film exposed to different atmospheres. **e** The current density-voltage characteristics for ZnMgO/Al film after different treatments. **f** Schematic of mechanism of the water-vapor treatment on the ZnMgO EIL.

species produced during the Al-water reactions at the porous ZnMgO-Al interface. Specifically, Supplementary Fig. 2 provides direct evidence through detection of both hydrogen molecules ($H_2$) and hydrogen radicals (H), substantiating this proposed mechanism. Although molecular hydrogen with moderate reductivity could transfer their electrons to the ZnMgO nanocrystals (Supplementary Fig. 3) followed by proton formation, its delayed detection relative to the onset of current and luminance enhancement (Supplementary Fig. 2e) and weaker reducing power compared to atomic hydrogen suggest $H_2$ plays a secondary role. Thus, we primarily attribute the doping mechanism to hydrogen radical-mediated electron transfer, though complete elucidation of their interfacial stabilization mechanisms awaits future study. The in situ n-doping of a portion to nearly all ZnMgO nanocrystals in the ZnMgO/Al film is quantively verified by ultraviolet-visible absorption spectroscopy (Fig. 1c). Al deposition alone induces ~20% bleaching of the ZnMgO's band-edge absorption, which intensifies to ~80% after the water-vapor treatment. This indicates nearly complete reductive-doping of the ZnMgO nanocrystals in the film (50% and 100% bleaching for one electron and two electrons per nanocrystal, respectively). In contrast, a pristine ZnMgO thin film exposed to water vapor shows negligible changes in the ultraviolet-visible absorption spectrum (Supplementary Fig. 4a). An oxygen treatment that removes the free electrons[27,30] subsequently extinguishes the intense EPR signal (Fig. 1b) and fully recovers the band-edge absorption (Fig. 1c). The restored band-edge absorption can be further bleached by a second round of water-vapor treatment (Supplementary Fig. 4b). These results collectively demonstrated that ZnMgO's n-type attributes are susceptible to oxidative defects[31–33] inevitably introduced during synthesis and processing, and the Al-water reaction is the critical factor in eliminating these defects and achieving heavy n-type doping.

Effective n-doping of ZnMgO nanocrystals within the ZnMgO/Al films should increase the carrier concentration, thereby elevating the Fermi level and lowers the surface contact potential of ZnMgO film (Fig. 1d). Though the contact potential of the pristine film remains largely unaltered by both water-vapor and oxygen treatments, it decreases by ~0.2 volts with just Al deposition and significantly by ~0.72 volts after the water-vapor treatment. Accordingly, the electron transport characteristic of the ZnMgO/Al film evolves from the space-charge limited mode to an ideal ohmic behavior due to the water-vapor treatment (Fig. 1e). Subsequently, an oxygen treatment reverts ZnMgO's surface contact potential to its original state (Fig. 1d), alongside a trap-filled limited current with a large power factor of voltage as 4.5 (Fig. 1e). This indicates the removal of doped free electrons and the regeneration of a significant number of electron traps. Quantitatively, conductivity of fully n-doped ZnMgO in the 0–100 mA cm$^{-2}$ range (~0.003 S m$^{-1}$) is on par with that of poly(3,4-ethylenedioxythiophene):poly(styrenesulfonate) (PEDOT:PSS, a mainstream hole-injection material used in QLEDs) (Supplementary Fig. 5a, b) and 3-6 orders of magnitude higher than those of the ZnMgO/Al film and the oxygen-treated ZnMgO/Al film (Fig. 1e). Consistent with spectroscopic analysis, an additional water-vapor treatment can bring the conductivity back to the state before the oxygen treatment (Supplementary Fig. 4c and Fig. 1e). It should be noted that, in addition to two overlapping curves both ended with the water-vapor treatment, the J-V measurements in Fig. 1e only include the ZnMgO/Al (the ZnMgO layer with the Al electrode) and /$H_2O$/$O_2$ (the sample after subsequent water-vapor and oxygen treatments).

The ZnMgO/Al film, rendered highly conductive by water-vapor treatment, exhibits excellent stability during on-shelf storage (Supplementary Fig. 5d). In contrast, the electron conductivity of the ZnMgO/Al film with acid-containing encapsulation peaks after a few days and then declines (Supplementary Fig. 5c). These observations suggest that the initial positive aging in QLEDs with acid-containing encapsulation[11,12,34] are likely due to n-doping of the EIL by reducing

hydrogen species produced from the reaction of volatile acids with metal electrodes.

Overall, the comprehensive data set from EPR, ultraviolet-visible absorption, surface contact potential, and electron transport measurements collectively confirm the in situ reductive-doping of ZnMgO nanocrystals in the film, as schemed in Fig. 1f.

## In situ reductive doping of electron-injection layer

The QLEDs presented here (Fig. 2a) feature a typical device structure, which includes a PEDOT:PSS hole-injection layer (HIL), an organic hole-transport layer (HTL), an emissive layer consisting of a monolayer of QDs[15,22,35] (Supplementary Fig. 6), a ZnMgO EIL/Al electrode, and acid-free encapsulation. Figure 2b shows that the band-edge absorption of the ZnMgO EIL in unencapsulated QLEDs follows the same response trend to Al deposition, subsequent water-vapor treatment, and subsequent oxygen treatment as observed in Fig. 1c. These results also indicate that successful n-doping of the ZnMgO EIL at the device level does not significantly impact the HTL layer. Additional experiments (Supplementary Fig. 7) confirm that direct water-vapor treatment on the HIL, HTL, and QD layers does not affect their optical or device optoelectronic properties. However, when the ZnMgO/Al film in QLEDs undergoes water-vapor treatment, there is a noticeable decrease in QD photoluminescence intensity, and its decay lifetime shortens from 6.2 to 4.3 ns, both of which can be reversed by the subsequent oxygen treatment (Fig. 2c). These findings suggest that, at the device level, reductive doping of the EIL greatly enhances electron transfer[30] between the heavily-doped ZnMgO EIL and the QDs, rather than increasing QD's radiative recombination quantum yield.

The substantially elevated free electron concentration and electron conductivity (Fig. 1) in the n-doped ZnMgO EIL facilitate seamless electron injection and transport across the EIL. After the water-vapor treatment with the Al electrode, the electron injection and transport of the ZnMgO EIL should be greatly improved according to the energy level alignments in Supplementary Fig. 3. Consequently, under external bias, the accumulation of free electrons in the QLEDs transitions from the ZnMgO-Al interface to the QD-ZnMgO interface. This shift is evidenced by an enhanced geometric capacitance that closely matches the expected capacitance contribution from the ZnMgO EIL, as illustrated in Fig. 2d and detailed in Methods.

The effective electron-injection barrier between the QDs and the EIL is determined by the energy disparity between the QD conduction band-edge and the ZnMgO Fermi level[36]. Figure 2e validates that, without water-vapor treatment, the current in the electron-only device (EOD) significantly trails that of the hole-only device (HOD), aligning with the low electron conductivity of the untreated ZnMgO/Al film (Fig. 1e). In accordance with the lifted Fermi level of the ZnMgO EIL (Fig. 1d), the water-vapor treatment markedly boosts the electron current, to an extent exceeding the hole current (Fig. 2e). This critical factor has been previously overlooked, presumably because EODs with acid-containing encapsulation exhibit strongly time-dependent electron currents (Supplementary Fig. 8a), similar to the ZnMgO/Al film with acid-containing encapsulation (Supplementary Fig. 5c).

Because of the spatial confinement of carriers within a QD, single electron injection in QLEDs provides an essential Coulombic attraction for subsequent hole injection, while repelling additional electrons. This leads to the reported mechanism for balanced charge injection and sustained high EQEs[9]. Thus, enhanced electron injection is expected to accelerate the electroluminescence (EL) cycle, boosting luminance at a low bias.

To delve into how water-vapor treatment modulates injection current and in turn, the device performance, we conduct an in situ investigation of water vapor's effect on unencapsulated QLEDs (Methods). Under continuous water vapor exposure, the current density at 3 V ($J_{3V}$) in a QLED gradually increases by one order of magnitude (Fig. 2f). The luminance at 3 V ($L_{3V}$) mirrors this trend yet

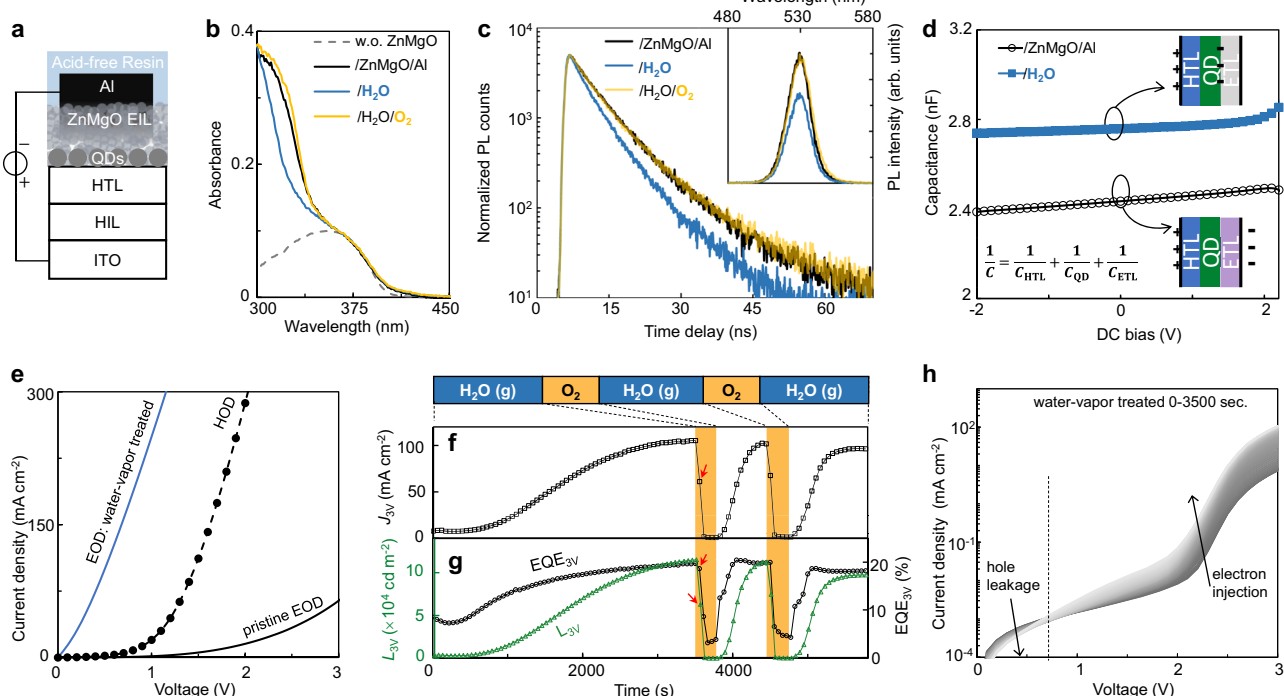

**Fig. 2 | Influence of n-type doping EIL on QLED device. a** Schematic of layers in the QLED structure. ITO indium-tin-oxide. **b** Ultraviolet-visible absorption spectra of QLED device after different treatments, with the ZnMgO-free device for reference (gray dashed line). **c** Transient and steady-state (inset) photoluminescence spectra of QD emitters embedded in QLEDs after different treatments. All photoluminescence measurements were performed on unbiased devices under open-circuit conditions. **d** The capacitance-voltage characteristics of a green-emitting QLED with different treatments. **e** Exposure to the water vapor markedly increases the current of EOD (with structure ITO/Polyethylenimine ethoxylated (PEIE)/QD/EIL/Al) to surpass the one of HOD (with structure ITO/HIL/HTL/QD/MoO$_x$/Au). **f, g** Temporal evolutions of in situ monitored luminance ($L_{3V}$), current density ($J_{3V}$), and EQE at 3 V of a green-emitting QLED exposed to the atmospheres indicated by the text-box above. Red arrows indicate slower degradation of EQE than current density upon oxygen treatment. **h** Current density-voltage characteristics for the device in e during the 1st-round water-vapor treatment.

shows a faster rate, leading to an early-reached EQE plateau (Fig. 2g). An oxygen treatment swiftly reverts the QLED to its original or even a more degraded state. Furthermore, an additional water-vapor treatment can bring the devices back to the state before the oxygen treatment (Fig. 2f,g and Supplementary Fig. 9), proving at the device level that the reductive treatment counteracts the adverse effects of oxidative species, whether accumulated during synthesis/processing or introduced via the oxygen treatment. This beneficial property is inherited from the ZnMgO/Al thin film (Fig. 1 and Supplementary Fig. 4b, c).

The presence of only ~1 monolayer of QDs between the HTL and EIL creates a potential pathway for direct hole leakage to the EIL trap states below the turn-on voltage[37–39]. Figure 2h reveals that during initial water-vapor treatment, current density exhibits a sharp increase above 0.6 V accompanied by a marked reduction below this voltage. This signifies that Al electrode-assisted in situ water-vapor treatment significantly enhances the ZnMgO EIL's dual functionality: improving electron transport efficiency while strengthening hole-blocking capability. Hole blockage and electron conductivity[40] are respectively associated with deep recombination centers and free-electron concentrations within the ZnMgO EIL schematically illustrated in Fig. 1f. During reductive treatment, the EQE increases more rapidly than the current density, whereas the oxygen treatment degrades the current density more quickly than the EQE, as indicated by red arrows in Fig. 2f,g. These effects are consistent with the higher energy of free electrons compared to that of the deep recombination centers within the bandgap.

The visible (450−550 nm) photoluminescence of ZnMgO nanocrystals, which is related to their deep-level recombination centers[28,32,33], and the ultraviolet (350−400 nm) emission from their intrinsic excitons[28,33,41], both respond actively to water-vapor and oxygen treatments (Supplementary Fig. 10a). Specifically, the water treatment with the Al electrode simultaneously suppresses the deep-level recombination and enhances the intrinsic exciton emission. The in situ reductive doping of the ZnMgO EIL also leads to a notable increase in both the open-circuit voltage and the short-circuit current of the QLEDs operating in photovoltaic mode (Supplementary Fig. 10b), supporting the reduction of the deep-level recombination centers and the enhancement of electron conductivity in the EIL.

These findings demonstrate that strategically incorporating electrons into the quantum-confined states at the conduction band edge of ZnMgO nanocrystals significantly enhances two critical processes: (1) electron transport within the EIL and (2) interfacial electron transfer from ZnMgO to QDs under forward-bias.

## Bright and efficient quantum-dot LEDs with n-doped layer

The state-of-the-art QLEDs reported in the literature, which utilize acid-containing encapsulation, fall short in simultaneously fulfilling all three key criteria (high IQE, high luminance, and low driving voltage), and they also exhibit significant positive aging effects (Supplementary Figs. 11–14). Positive aging describes the QLED's performance improvement during storage, on-and-off cycling, and high-current stress, which occurs unpredictably and is not reproducible within a timeframe of days. The carboxylic acids from the resin slowly and irregularly infiltrate the EIL-metal interface[12], a process hindered by their larger molecular size, and react with the electrodes to generate the reducing hydrogen species necessary for effective n-doping. This hypothesis is corroborated by the observed variations in luminance uniformity throughout the shelf-aging process (Supplementary Figs. 11–14, photographs). Considerable research efforts are being invested to address these effects[11,12,33,34,37,42,43] (Supplementary Note 1).

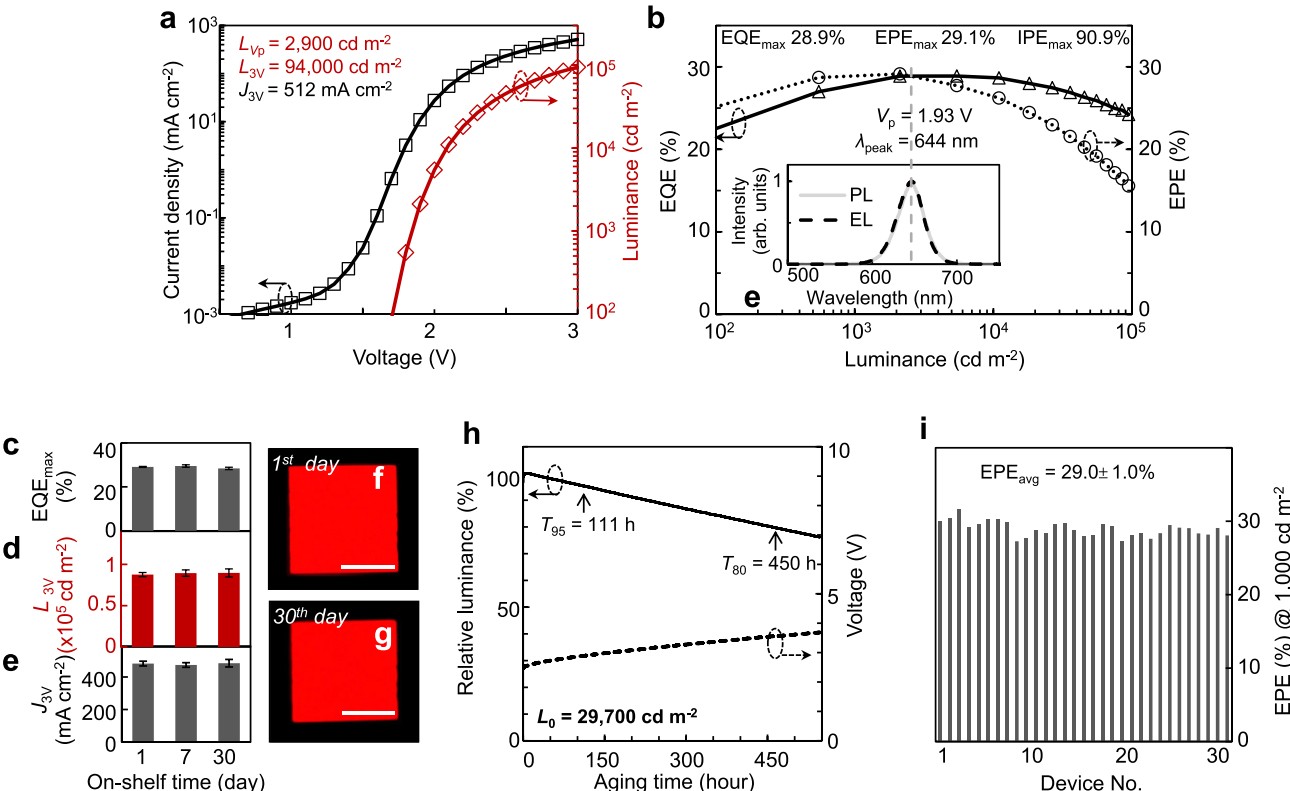

**Fig. 3 | Bright QLEDs with near-unity internal power efficiency. a** Current density-voltage-luminance (*J*-*V*-*L*) characteristics of a red-emitting (644-nm) QLED. **b** EQE and EPE versus luminance for the device shown in (**a**) in a broad luminance range. Inset: photoluminescence (PL, solid line) of the QD solution and electroluminescence (EL, dashed line) of a QLED. The crossing point of EPE and EQE curves corresponds to the average energy of emitted photons (1.93 eV). **c**–**e** Peak EQE ($EQE_{max}$), luminance ($L_{3V}$) and current density ($J_{3V}$) at 3 V versus shelf-aging time. Error bars represent the standard deviations derived from one batch of four typical devices. **f**, **g** the photographs of a working QLED on the 1st and 30th day (scale bar: 1 mm). **h** Relative luminance and voltage versus operational time (driven at 150 mA cm⁻²). **i** EPE at 1000 cd m⁻² from 31 devices.

QLEDs with acid-free encapsulation exhibit minimal positive aging effects, yet their EQE and luminance are inferior, particularly pronounced in high-energy photon-emitting devices (Supplementary Figs. 11–14). Through systematic comparisons between QLEDs of different emission colors and EODs with acid-containing and acid-free encapsulation (Supplementary Fig. 8), we hypothesize that the inferior performance—low EQE and luminance—of the devices with acid-free encapsulation can be traced back to the scarcity of free electrons and abundance of trap-assisted recombination centers within the ZnMgO EIL. In principle, the reductive treatment efficiently and controllably maximizes all necessary enhancements on the electron injection and transport of the ZnMgO ETL prior to completion of the device fabrication by the acid-free encapsulation, which also pre-excludes all negative aging effects caused by the acidic resins in the relatively long term (Supplementary Figs. 11–14). Given the inherent limitations of pristine ZnMgO EILs, specifically their inadequate electron conductivity and subpar hole-blocking capabilities, we anticipate a universal enhancement trend in water-vapor treated QLEDs spanning all emission colors (see detail below). Moreover, a truly n-doped EIL should largely boost the luminance at low driving voltages while maintain an ideal charge balance[9], which is particularly crucial for QLEDs—such as green and blue ones—facing electron-injection challenges.

Operating at $V_p$ (1.93 V), a typical red-emitting (644-nm) QLED with the in situ n-doped EIL achieves a luminance up to 2900 cd m⁻² and a peak EQE of 28.9% (Fig. 3a, b). At a driving voltage of 3 V, the luminance soars to 94,000 cd m⁻². With an LEE of 32% (Supplementary Fig. 6), the internal power efficiency (IPE, calculated as EPE/LEE) is

maintained above 81% across a luminance range of 500–10,000 cd m⁻² (peak IPE 90%). Our devices exhibit high operational- and shelf-stabilities, in contrast to the positive aging effects observed in QLEDs with acid-containing encapsulation. Post shelf-storage for varied periods, the QLEDs maintain nearly constant EQE, luminance, current density, emission uniformity (Fig. 3c–g). Stressed at a constant current density (150 mA cm⁻²) with a high initial luminance (~30,000 cd m⁻²), the device demonstrates long $T_{95}$ (111 h) and $T_{80}$ (450 h) lifetimes without positive aging (Fig. 3h), meeting the requirements for general lighting. The devices fabricated with the proposed scheme are reproduceable, with a statistical IPE of 90.6 ± 3.1% at 1000 cd m⁻² for a batch of 31 red-emitting (644-nm) QLEDs (Fig. 3i).

The U.S. Department of Energy (DOE) 2022 Solid-State Lighting R&D Report identifies operational voltage minimization as a pivotal strategy for achieving energy-efficient general lighting, setting 2035 performance targets at luminance of 8000 cd m⁻² at 3.0 V bias[44]. Though challenging, achieving operating voltages near or below $V_p$ would be ideal. The reductive treatments introduced here are readily extended to QLEDs across the visible spectrum, enabling low-voltage operation with high efficiency. Experimental results reveal 2.6–12.3 times luminance enhancement at $V_p$ compared to conventional devices (Fig. 4a–c, Supplementary Table 1 and Figs. 12–15). As a result, the luminance level at 3 V substantially exceeds the DOE targets across the visible spectrum (Fig. 4a–c). Notably, these QLEDs exhibit exponential luminance escalation when driven at ~1.5 times of $V_p$ (Fig. 4d), reaching application-ready brightness for lighting/display systems without requiring optical extraction enhancements. As anticipated, the higher the emitting photon energy, the greater the improvements observed

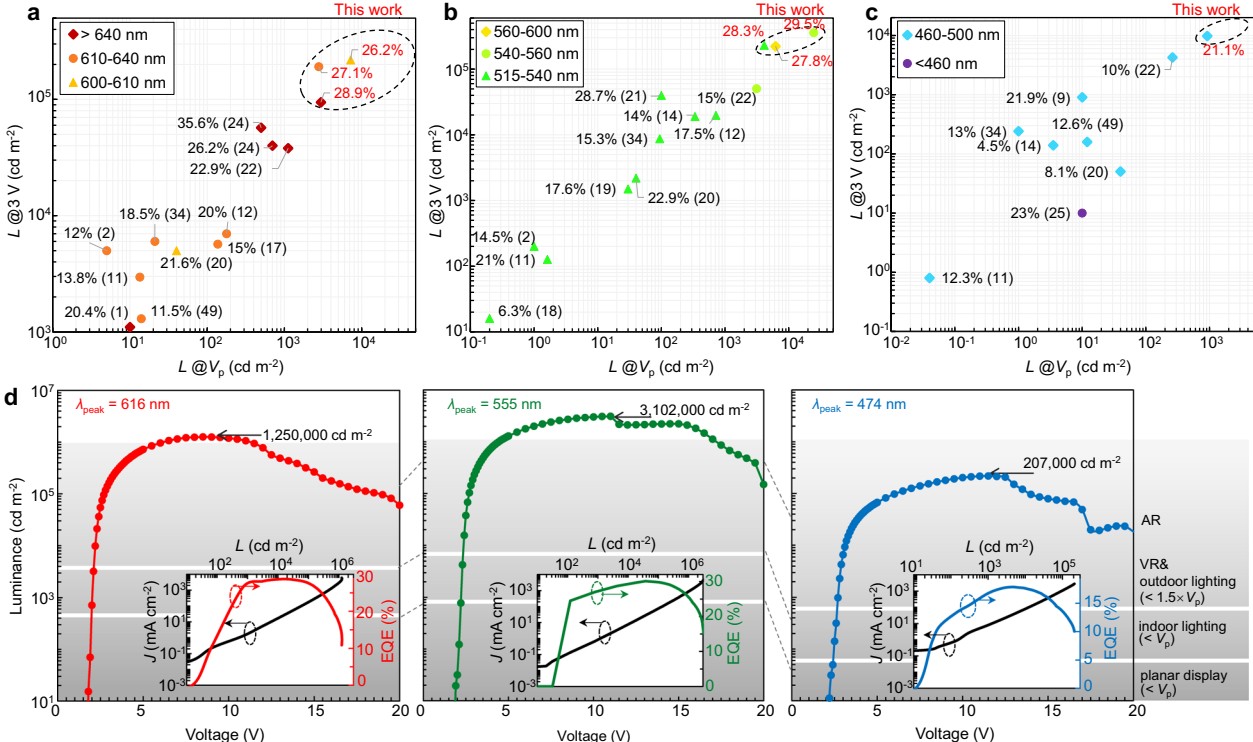

**Fig. 4 | Bright and energy-efficient QLEDs across the visible spectrum.** Comparative analysis of our top-performing QLEDs against previously reported high-performing QLEDs in the orange-red (**a**), green-yellow (**b**), and blue-cyan (**c**) spectral windows, respectively. The x- and y-axis in each plot is the luminance at the corresponding $V_p$ and at a 3 V bias. The peak EQE values are annotated next to each data point, with the respective reference numbers in parentheses. **d** Voltage-luminance characteristics of orange-red-, green- and blue-emitting QLEDs under intense electrical excitation. The shaded areas indicate the luminance levels required for different scenarios outlined to the right. Inset, current density-luminance-EQE characteristics under DC bias.

with the n-doped nanocrystalline EIL (Supplementary Fig. 16), suggesting that the performance of the green and blue QLEDs based on state-of-the-art techniques is indeed limited by suboptimal electron injection.

Our QLEDs across the visible spectrum reaches unpresented maximum luminance (Fig. 4d, red: 1,250,000, green: 3,102,000, and blue: 207,000 cd m$^{-2}$), limited by electron leakage[26] or thermal quenching effects. Notably, the driving voltages at the record-high brightness for all colors are quite low (only ~10 V). With pulsed excitation to mitigate thermal stress[6], transient radiance levels during the excitation period achieve unprecedented high values (red: 42,320, green: 35,051, and blue: 16,713 W sr$^{-1}$ m$^{-2}$, Supplementary Fig. 17), showcasing the full potential of QLEDs under high excitation levels when thermal dissipation is not a limiting factor. These excitation levels approach the population-inversion threshold necessary for realizing QD laser diodes[6]. For instance, in red-emitting QLEDs, a distinct high-energy emission peak emerges when the radiance exceeds ~20,136 W sr$^{-1}$ m$^{-2}$ under 9 V bias, exhibiting spectral characteristics remarkably similar to those induced by pulsed laser excitation (Supplementary Fig. 18). This phenomenon can be attributed to the recombination process involving $p$-shell electrons, suggesting the occurrence of population inversion in the $s$-shell energy states[6]. We expect that the performance of our device would be further boosted via enhanced hole injection and blocked electron leakage[26].

The in situ n-doping strategy proves to be consistently effective for the cadmium-free QLEDs (Supplementary Fig. 19) and EILs synthesized through various routes (Supplementary Fig. 20). While our methodology development primarily centers on the ZnMgO systems, the demonstrated universality is evidenced by enhanced performance in ZnO-based QLEDs (Supplementary Fig. 20). Its efficacy is similarly robust when

applied to QLEDs with either titanium or aluminum electrodes (Supplementary Fig. 21a−c). However, the scenario is slightly different for QLEDs featuring inert metal electrodes, such as silver and gold. According to the electrode potentials shown in Supplementary Fig. 3, reduction potentials of the Ag/Ag$^+$ and Au/Au$^+$ (or Au/Au$^{3+}$) redox pairs are too low to initiate the hydrogen reduction reaction. The performance of these QLEDs shows notable improvement during continuous $J$-$V$ sweeps (Supplementary Fig. 21d−i)−another kind of positive aging reported in the literature[43]−which we attribute to the electrochemical reduction of water at the inert electrode[45] under forward bias (Supplementary Note 2).

**Broadband quantum-dot LEDs and white-lighting**
QLEDs with a stably and heavily doped EIL simultaneously satisfy all three key parameters for general lighting: optimal IQE, high luminance, and operation at sub-bandgap driving voltage. However, high-quality general lighting demands a nearly continuous emission spectrum, which would greatly benefit from sub-bandgap-voltage-driven broadband QLEDs ($\beta$QLEDs). These $\beta$QLEDs are fabricated using a monolayer blend of QDs with slightly different emitting photon energies[46]. Figure 5a demonstrates that, at a practical current density (1 mA cm$^{-2}$) for general lighting, driving voltage of the QLEDs is linearly related to and ~0.15 V below its $V_p$, indicating near-ideal electron injection from the in situ n-doped EIL across the visible spectrum.

A prototype red $\beta$QLED, utilizing a blend of QDs emitting at 605, 623, and 644 nm and operating at 1.90 V−below the average photon voltage−yields EL with a 55 nm full-width-half-maximum, a luminance of 1050 cd m$^{-2}$, and an impressive IPE of 91.9% (EPE of 29.4% without light extraction) (Fig. 5b). This $\beta$QLED exhibits minimal color shift over a wide luminance range and negligible color drift after 100-h

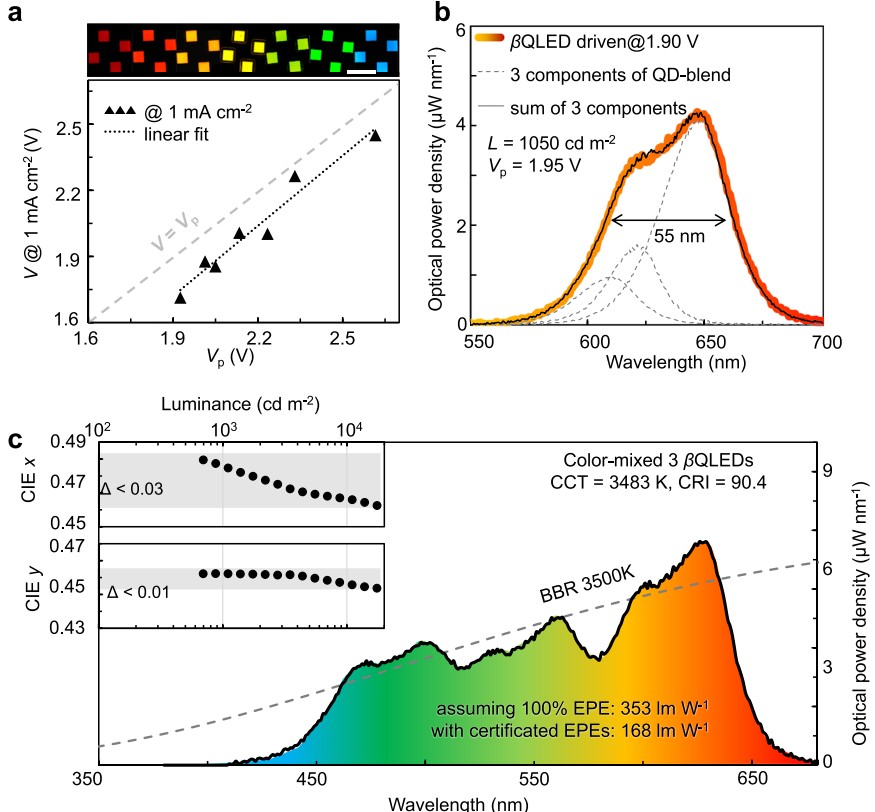

**Fig. 5 | Broadband QLED and proposed white-light QLED. a** Summary of correlation between $V_p$ of QDs in QLEDs and operating voltages of the devices at current density of $1\,mA\,cm^{-2}$ ($V@1\,mA\,cm^{-2}$). Top: the photographs of working QLEDs emitting different colors with a scale bar of 5 mm. **b** Electroluminescence spectra of an orange-red $\beta$QLED produced using a blend of three types of QDs. Dashed lines show individual contributions from the three types of QDs comprising the emissive blend. $V_p$ represents the average photon energy calculated through spectral intensity-weighted integration of the electroluminescence profile, accounting for the contributions from all QD components in a $\beta$QLED. **c** Proposed color-mixed white-light QLED composed of three $\beta$QLEDs achieving a CRI of 90.4 referring to the black-body-radiation (BBR) at 3500 K. Inset, CIE-$x$ and $y$ values of the color-mixed $\beta$QLEDs as a function of luminance.

continuous operation at a constant voltage (Supplementary Fig. 22a–c). Green and blue $\beta$QLEDs, each incorporating a blend of three kinds of QDs, are also fabricated and present acceptable color shift (Supplementary Fig. 22d–e). The combination of these three $\beta$QLEDs enables the creation of a color-mixed white-light source with minimum color shift across a luminance range of 600–20,000 cd m$^{-2}$ (Fig. 5c inset and Methods). It is anticipated that, by further improving the hole injection[26], the driving voltage at the given luminance range (600–20,000 cd m$^{-2}$) could be reduced for achieving IPE significantly over unity.

We propose a white-light source fabricated through an idealized spectral combination of $\beta$QLEDs, closely emulating the black-body radiation at 3500 K, achieving an excellent color-rendering index of 90 (Fig. 5c and "Methods"). With the certified EPEs for the monochromatic QLEDs coupled with semispherical lenses (55%, 46%, and 32% for red, green and blue ones, respectively, Supplementary Fig. 23), luminous efficacy of the resulting warm-white QLED is calculated to be 168–181 lm W$^{-1}$ within an intensity range of 5500–55,000 lm m$^{-2}$, corresponding to an overall EPE of ~50% (Supplementary Fig. 22f and "Methods"). This efficacy and CRI at high luminance surpass the R&D goals for diffuse light sources set by the U.S. Department of Energy for the year 2035 (CRI ≥ 80 and efficacy ≥ 150 lm W$^{-1}$ at 10,000 lm m$^{-2}$)[44].

## Discussion

A straightforward water-vapor treatment of QLEDs with a top metal electrode enables in situ n-type doping of the ZnMgO EIL, leading to ideal electron injection and hole blockage capabilities that surpass those of conventional organic EILs[47] and untreated ZnMgO EILs. QLEDs

with the heavily n-doped nanocrystalline EIL simultaneously match three key parameters needed for high-quality general lighting. The exceptional conductivity of n-doped nanocrystalline EIL and the mechanisms behind open up promising opportunities for next-generation electronic and optoelectronic devices.

## Methods
### Chemicals

Tetramethylammonium hydroxide pentahydrate (TMAH, 98.5%) and ethanol (extra dry, 99.5%) were purchased from J&K Chemical. Zinc acetate dihydrate (99.99%), magnesium acetate tetrahydrate (99.98%), 2-(2-(2-methoxyethoxy)ethoxy)acetic acid (80%), dimethyl sulfoxide (DMSO, 99.8%), octane (GC grade) and tetrahydrofuran (THF, 99.99%, no stabilizer) were purchase from Aladdin Industrial Corp. Ethyl acetate (99.5%) and methanol (MeOH, 99.5%) were purchased from Sinopharm Chemical Reagent. Chlorobenzene (extra dry, 99.8%) and 5,5-dimethyl-1-pyrroline N-oxide (DMPO) was purchased from Sigma-Aldrich. Poly(ethylene-dioxythiophene):polystyrene sulfonate (PEDOT:PSS) was purchased from Heraeus. Poly((9,9-dioctylfluorenyl-2,7-diyl)-co-(4,4'-(N-(4-sec-butylphenyl)diphenylamine))) (TFB) and poly((9,9-dioctylfluorenyl-2,7-diyl)-alt-(9-(2-ethylhexyl)-carbazole-3,6-diyl)) (PF8Cz) were purchased from Volt-Amp Optoelectronics Tech. Co., Ltd, Dongguan, China. Triphenylmethylium tetrakis-(penta-fluorophenyl-borate) (Tr-TPFB) was purchased from TCI Chemical. Polyethylenimine ethoxylated (PEIE) was purchased from Hwrkchemical Co., Ltd, Beijing, China. Colloidal QDs used in this work were purchased from Najing Technology. All chemicals were used directly without any further purification unless otherwise stated.

## Synthesis of zinc oxide nanoparticles

The synthesis of ZnO nanoparticles was modified from a previously reported method[1]. In brief, a solution of zinc acetate dihydrate (3 mmol) in DMSO (30 ml) was mixed with a solution of 2-(2-(2-methoxyethoxy)ethoxy)acetic acid (3 mmol) and TMAH (7.5 mmol) in ethanol (10 ml). The mixture was stirred for 1 h at room temperature before undergoing purification using the same procedures as ZnMgO (see below). The resulting oxide nanocrystals were then redispersed in ethanol and filtered (0.22 μm Nylon66 filter) before use.

## Synthesis of magnesium-modified zinc oxide nanoparticles

The synthesis of ZnMgO nanoparticles was modified from a published method[19]. In brief, for ZnMgO synthesized with TMAH, a solution of zinc acetate dihydrate (2.55 mmol) and magnesium acetate tetrahydrate (0.45 mmol) in DMSO (30 ml) was mixed with a solution of TMAH (4.5 mmol) in ethanol (10 ml). The mixture was stirred for 1 h at 50 °C, followed by purification through repeated dispersion and precipitating using ethanol and ethyl acetate, respectively. The resulting oxide nanocrystals were redispersed in ethanol and filtered through a 0.22 μm Nylon66 filter before use. For ZnMgO synthesized with KOH, TMAH was substituted with KOH (in equal molar amount), and the other steps of the recipe remains the same.

## Fabrication of quantum-dot light-emitting diodes

For the red-emitting (644-nm) device, prepatterned indium tin oxide (ITO)-coated glass substrates, with a square resistance of 10 Ohm sq$^{-1}$, were subjected to a cleaning regimen in an ultrasonic bath using acetone, deionized water, and ethanol. This was followed by a 15-min air plasma treatment utilizing a plasma cleaner (Harrick Plasma, PDC-002). Subsequently, PEDOT:PSS solutions (Al 4083, filtered through a 0.22-μm PVDF filter) were spin-coated onto the ITO substrates at 3000 revolutions per minute (r.p.m.) for 45 s, then baked at 150 °C for 30 min in air, prior to being transferred into a nitrogen-filled glovebox ($O_2 < 1$ ppm, $H_2O < 1$ ppm). The TFB (in chlorobenzene, 8 mg ml$^{-1}$, with Tr-TPFB doping to enhance hole injection[48]), quantum dots (QDs, in octane, ~15 mg ml$^{-1}$), and ZnMgO nanoparticles (in ethanol, ~23 mg ml$^{-1}$) were layer-by-layer deposited by spin coating at 2,000 r.p.m. (for TFB and ZnMgO) or 2500 r.p.m. (for QDs) for 45 s each. The TFB layers were baked at 150 °C for 30 min before the application of the QD layers. Metal electrodes, consisting of 70 nm aluminum, 70 nm titanium, 70 nm silver or 70 nm gold, were then vacuum-deposited onto the ZnMgO layers through a shadow mask, using a thermal evaporation system (QHV-R96, Shenyang Qihui Vacuum Technology Co., Ltd) under a base pressure of $5 \times 10^{-4}$ Pa. The active device area was 4 mm$^2$, defined by the overlap area of the ITO anode and the metal cathode.

For devices treated with different atmospheres, they were placed onto a rack after metal deposition, then sealed in an atmosphere-controlled chamber for a designated duration. During this time, different atmospheres passed through the chamber at a fixed flow rate whose humidity is controlled by a home-built system (described below). Finally, acid-free UV-curable resins (LOCTITE 3335 unless otherwise stated) were used to encapsulate the devices by covering glass slides in a glovebox.

For 532- and 474-nm QLEDs, PF8Cz was used as HTL instead of TFB and the specific thickness of each layer are detailed in Supplementary Fig. 6. For other colors, layer thicknesses were meticulously optimized based on LEE calculations.

For the devices operating under pulsed electrical excitation, a current-focusing structure was introduced. The fabrication procedure deviated from the standard one only by incorporating a 50-nm-thick MgF$_2$ insulating layer, inserted between the ZnMgO and metal cathode via vacuum thermal evaporation using a shadow mask which defined a 100-μm-wide slit. The top Al-cathode stripe was deposited using a 200-μm-wide shadow mask orthogonal to the slit defined by MgF$_2$. By confining the current injection to an area of $100 \times 200$ μm$^2$ and

operating in pulsed mode, the device significantly mitigated the thermal stress typically caused by large current densities.

## Characterization of quantum dots

Absorption spectra were obtained using an Agilent Cary 4000 Ultraviolet–visible spectrophotometer, with the slit set to a 2-nm bandwidth. Photoluminescence (PL) spectra were captured with an Agilent Cary Eclipse fluorescence spectrophotometer. Red-, green- and blue-emitting QDs were excited at 400 nm, 380 nm, and 300 nm, respectively, with the excitation slit width set to 10 nm. The width of the emission slit was adjusted to 1.5 nm for red and green QDs and 2.5 nm for blue QDs. For transmission electron microscopy (TEM) measurements, QDs were deposited onto a copper grid with an ultrathin carbon film. TEM images were then acquired using a Hitachi 7800 operating at an acceleration voltage at 100 kV.

## Setup for water-vapor treatment

An apparatus was designed to generate argon (Ar) gas with a precise humidity level. The system's operation was managed by two mass flow controllers that are responsible for delivering dry Ar and moisture-saturated Ar through a gas-washing bottle, respectively. These controllers meticulously regulated the flow rates, with the dry Ar flow typically set to 360 milliliters per minute (ml min$^{-1}$), and the moist Ar flow to 240 ml min$^{-1}$. The Ar streams merged in a buffer tank, where they mixed to achieve the targeted humidity level, as monitored by an integrated humidity sensor. The resulting humidified Ar gas was then directed into the sample chamber. After a predetermined duration, the sample chamber was desiccated with dry Ar gas and transferred back to the glove box.

For oxygen treatment, the air flow ratio of pure oxygen or a mixture containing 95% nitrogen and 5% oxygen to pure nitrogen was set at 1:4 (mixture gas flow rate: 50 ml min$^{-1}$, nitrogen flow rate: 200 ml min$^{-1}$).

## Integrating sphere-based *J-V-L* characterization

The current density–voltage–luminance (*J-V-L*) characteristics and EL spectra were measured with a home-made system consisting of a digital source meter (Keithley 2400 or Keithley 2450) and an integrating sphere (ISP-50, Ocean Optics) connected to a spectrometer (QEPro, Ocean Optics). Absolute spectral radiant flux calibration was executed with a traceable radiant-flux standard lamp (HL-3 plus, Ocean Optics). The standard spectral luminous efficiency function (CIE 1931 V(λ)) was applied for determining the photometric quantities. All electrical sweep and data acquisition procedures were automated and controlled by a computer via Python scripts. The voltage sweep increment was set to 0.1 V and integration time of spectrometer was set to 20 ms unless specified otherwise.

For the electrical-pulse excitation setup, the source meter was set to direct current (DC) voltage mode and connected to the anode of the QLED under test. The cathode of the QLED was attached to the collector port of a bipolar junction transistor (BJT 2N3904). The base of this transistor was controlled by a microcontroller unit, which was responsible for modulating the electrical pulses. The emitter port of the transistor was connected to a sampling resistor to measure the current. The microcontroller unit was programmed to generate electrical pulses with a duration of 5 μs and a repetition rate of 250 Hz as depicted in Supplementary Fig. 17. The temporal response of QLED under electrical pulse excitation was captured by a photodiode (Thorlabs PDA100A2 with the gain set to zero). To access the luminance integrated over time, the integrating sphere coupled with a spectrometer, as previously described was utilized. The raw time-integrated (integration time 4000 ms for each voltage point) data obtained from the spectrometer were then adjusted to reflect the actual radiance and luminance during the active phase of the electrical pulse. This was achieved by multiplying the measured luminance by the reciprocal of the duty cycle (1/800). The duty cycle is the ratio of

the pulse width to the total period of the pulse train, and in this case, it accounts for the fraction of time the QLED is actively being excited.

## Operational lifetime measurement

The operational lifetimes of QLEDs were measured under ambient conditions using a commercial LED-lifetime test system either from Crysco Equipment (Guangzhou) or EVERFINE Corporation (Hangzhou). The operational lifetime curves presented are raw data without any manipulation. Prior to the formal lifetime measurements, the QLEDs were subjected to bias stress at a constant current density identical to the aging current density for 5 min.

## In situ *J-V-L* characterization of quantum-dot LEDs

The in situ current density−voltage−luminance (*J-V-L*) curves were obtained using a home-built system comprising a source meter (Keithley 2450) and an optical imaging system equipped with a monochrome CMOS camera (MER-301-125U3M, DAHENG IMAGING).

Following the deposition of aluminum cathodes, the QLED device was transferred to a sample chamber equipped with an optical window. Subsequently, the sample chamber was connected to the humidity control system integrated with the home-built *J-V-L* characterization system. During water-vapor treatment, the device was subjected to a DC bias with a sweep voltage range of 0–3 V (in step of 0.1 V, with each cycle lasting around 5 s). Subsequently, a monochrome CMOS camera captured images when the device was driven by a 3-V DC bias, a 10 mA cm$^{-2}$ DC current and under open-circuit condition excited by a program-controlled 355 nm LED successively. The imaging session lasted 5 s. The source meter was then switched off for 50 s before the next round of *J-V-L* measurement cycle. The time interval for each *J-V-L* measurement cycle was ~60 s. All procedures were automated and controlled by a computer via Python scripts.

All the raw counts recorded by CMOS camera are within the linear range. The CMOS camera's counts of each pixel were summed to compute the relative luminance at a 3-V bias ($L_{3V}$). The ratio of $L_{3V}$ to $J_{3V}$ was calculated as the relative external quantum efficiency ($EQE_{3V}$). The relative $L_{3V}$ and $EQE_{3V}$ were further converted to absolute values using a factor derived from integrating sphere-based *J-V-L* measurements once the in situ characterization was completed.

## Steady state photoluminescence of ZnMgO films

PL spectra of ZnMgO films were captured using an Agilent Cary Eclipse fluorescence spectrophotometer. ZnMgO films were excited at 300 nm, the width of excitation and emission slits were both set as 1 nm. To eliminate the atmospheric effects and ensure that measurement results were not influenced by UV-curable resins, the ZnMgO films were encapsulated using a special cover glass, featuring a pre-etched central region that create a hollow space above the region of interest.

## Ultraviolet−vis absorption spectra of ZnMgO films

Absorption spectra of ZnMgO films were obtained using an Agilent Cary 5000 ultraviolet−visible-near-infrared spectrophotometer equipped with an internal diffuse reflectance accessory (DRA-2500). For absorption measurement, the sample was fixed at the exit port of the integration sphere of DRA-2500, and a quartz substrate coated with thermally evaporated aluminum was used as reference.

The ZnMgO nanocrystals (~23 mg mL$^{-1}$) was spin coated on a quartz substrate, on top of which a 70-nm aluminum film was deposited via thermal evaporation. Subsequently, the film was treated with different atmospheres. Finally, the sample was encapsulated using a special cover glass, featuring a pre-etched central region that create a hollow space above the region of interest.

## Surface potential measurement

The surface potential measurements were performed under nitrogen conditions with a Single-Point Kelvin Probe system (KP020, KP

Technology). Each data point was averaged from 40 measurements and the error values tested were less than 40 mV. The ZnMgO nanocrystals (~23 mg mL$^{-1}$) were spin coated on ITO substrate, and then exposed to different atmospheres (water-vapor treatment for 40 min and oxygen-treatment for 60 min after water-vapor treatment, respectively). For the ZnMgO/Al film, the aluminum film was removed after treatment and before the measurement using a scotch tape.

## Electron paramagnetic resonance characterizations

Electron paramagnetic resonance (EPR) spectra were recorded at room temperature using a Bruker A300 EPR Spectrometer, operating at an X-band frequency of ~9.8 GHz. To simulate the thin-film state, ZnMgO was drop-coated onto glass substrates and then carefully scraped from the substrates, followed by transferred into a quartz EPR standard quality tube with an outer diameter of 5 mm. The ZnMgO sample was first treated with water vapor for 60 min, followed by UV irradiation for 30 h.

For the ZnMgO/Al film, ZnMgO was first drop-coated onto glass, followed by the evaporation of Al. The coated glass was then carefully scraped to retrieve the ZnMgO/Al film, which was then transferred to a 5-mm diameter quartz tube. The sample underwent sequential treatments: first exposed to water vapor for 30 min, then to oxygen for 90 min. EPR spectra were recorded for both samples in their pristine state and subsequent to each treatment.

To detect the hydrogen radical, the evaporated Al was scraped from glass substrates. Following a 4-h water vapor treatment, the scraped Al was then dispersed into a DMPO diluent (volume ratio of 1:10 in THF) for 5 min. The DMPO diluent, following the reaction, was then transferred into a glass capillary and sealed with silicone grease. For the blank sample, the DMPO diluent was transferred directly into a glass capillary and sealed with silicone grease, without any exposure to Al. Subsequently, the glass capillaries were inserted into a quartz tube for EPR analysis. All sample preparations were conducted under a nitrogen atmosphere.

## Gas chromatography

To ascertain the gaseous byproducts of the ZnMgO/Al film following an excess water-vapor treatment, it was necessary to enhance the yield of gaseous species. The samples were fabricated via spin coating of ZnMgO, followed by the evaporation of aluminum. Thirty film samples were sealed in a nitrogen-filled sample chamber which has a capacity of ~200 mL, along with a dust-free cloth imbued with water. The chamber was subjected to a temperature of 40 °C to facilitate and accelerate the reaction kinetics. After 2 h, an additional 20 mL of air was introduced into the chamber to displace and purge the gaseous product, which were then injected into a gas chromatograph equipped with a thermal conductivity detector (Agilent, GC-7890B). Two standard gas mixtures with concentrations of $H_2/N_2$ (100 ppm and 5%) was used to ensure accurate quantification and identification of the hydrogen gas produced in the reaction.

## Cross-sectional characterizations of quantum-dot LEDs

The cross-sectional samples used for transmission electron microscope (TEM) characterizations were firstly fabricated on a silicon substrate. Subsequently, they were prepared using a dual-beam focused-ion-beam system (ThermoFisher SCIENTIFIC Helios 5 UX). The high-angle annular dark-field (HAADF) images and elemental mappings were recorded using a transmission electron microscope (ThermoFisher SCIENTIFIC Talos F200X G2).

## Capacitance−voltage characterization

The capacitance−voltage (C-V) measurements were carried out with a Keysight E4980A precision LCR meter and the data was automatically acquired by a computer via Python scripts. To minimize the impact of trapped charges, a modulating frequency of 10 kHz and a modulating

amplitude of 10 mV were employed. As shown in Fig. 2d, the capacitance value of QLED stay almost constant up to a bias of 2 V, reflecting the geometric capacitance contributions from the HTL, QD layer, and EIL, expressed as $\frac{1}{C} = \frac{1}{C_{HTL}} + \frac{1}{C_{QD}} + \frac{1}{C_{EIL}}$. However, the device treated with water-vapor after Al deposition exhibits an enhanced geometric capacitance. This is attributed to the passivation of trap states and the reactivation of the n-type characteristics of the ZnMgO layer. These effects collectively facilitate electron injection and transport across the ZnMgO layer with minimal energy barriers, resulting in electron accumulation at the interface between the QDs and ZnMgO.

The calculated differences in reciprocal capacitance for devices with and without water-vapor treatment are $(23\,nF)^{-1}$, $(21\,nF)^{-1}$, and $(21\,nF)^{-1}$ for red-, green-, and blue-emitting QLEDs, respectively. These values are in close proximity to the geometric capacitance of ZnMgO, which is calculated to be 18 nF, based on literature parameters[49]. Consequently, the ZnMgO layer in devices treated with water vapor should not be perceived as a depletion layer; instead, the capacitance of QLED is primarily determined by the geometric capacitance of the HTL and the QD layer.

### Light extraction efficiency calculation

The calculation of light extraction efficiency was performed following the method developed by K. A. Neyts[50]. In brief, the QD was treated as an electrical dipole antenna situated at the center of the emissive layer, with random orientation. The multi-layer structures on the top and bottom of the emissive layer were regarded as mirrors of a microcavity, with their reflection and transmission coefficients determined through an iterative computational process. The power density radiated was decomposed according to the in-plane wave vector ($k_p$). All complex indices were measured independently using an ellipsometer (HORIBA UVISEL). The physical thickness of functional layers was measured using a step profiler (Bruker DektakXT) or an atomic force microscopy (Park NX10). To eliminate errors induced by different substrates, the thickness was initially measured for the layer-of-interest along with its underlying layer, followed by the measurement of the underlying layer alone. The difference between these two measurements provided the precise thickness of the layer-of-interest.

### Color coordinate of color-mixed white-light quantum-dot LED

The color shift of red, green and blue (RGB) $\beta$QLED across different luminance range was calculated based on the spectral energy distribution measured at different bias voltages and the Commission Internationale de l´Eclairage (CIE) color matching functions for the standard observer. For a hybrid luminaire comprising RGB $\beta$QLEDs, the calculation of color shift was methodically approached by first quantifying the luminance output of each color component. Subsequently, the color coordinates for each $\beta$QLED were identified, reflective of their spectral attributes at the specified luminance. The final step consolidated these data points, deriving the composite color coordinates through a weighted average that corresponded to the proportional luminance of the RGB components. This methodology ensured a precise representation of the hybrid luminaire's color shift within the desired luminance range.

### Hybrid quantum-dot LED for general lighting

In the main text, we detailed the creation of a red broadband QLED ($\beta$QLED) by blending three varieties of red QDs. To extend this technology for white lighting, we conceptually shifted the spectrum of the red $\beta$QLED to cover the orange-red, green-yellow, and blue-green regions. Under the constraints of a correlated color temperature (CCT) within $3500 \pm 50$ K and a color rendering index (CRI) above 90, we optimized the relative power density and central wavelengths of these three spectral components. This optimization led to a maximum attainable efficacy, as depicted in Fig. 5c and detailed in Supplementary Fig. 22f. The maximum efficacy (Efficacy$_{max}$) was calculated under

the assumption of 100% EPE, using the following formula:

$$\text{Efficacy}_{max} = 0.466 \times 312 + 0.281 \times 596 + 0.253 \times 159 = 353\,\text{lm W}^{-1} \qquad (1)$$

To gauge the efficacy of the envisioned color-mixed white QLED in its current stage, we assumed that a total light output of 4000 lumens should be provided by a $60 \times 120$ cm² flat panel troffer. Based on the optimized energy distribution, the lumen contributions of the orange-red, green-yellow and blue-green parts can be calculated as 29%, 57%, and 14%, respectively. Assuming equal emissive areas for each color component and a Lambertian emission pattern, the corresponding luminance can be estimated as 1539, 3025, and 743 cd m$^{-2}$ for the red-, green-, and blue-emitting component, respectively. These luminance levels are achievable under a bias at the voltage equivalent to the bandgap voltage. For the sake of simplicity in our subsequent calculations, we have assumed that the orange-red, green-yellow, and green-blue $\beta$QLEDs all operate at a voltage equal to the average photon voltage. By applying light extraction through the use of a hemispherical lens, the certified EPE (measured by China Electronic Product Reliability and Environmental Testing Research Institute, CEPREI) can at least reach 32%, 46%, and 55% for the blue-, green-, and red-QLED (Supplementary Fig. 23). As a result, we can estimate the real efficacy based on our devices via the equation:

$$\begin{aligned}\text{Efficacy}_{real} &= 0.402 \times 312 \times 0.55 + 0.290 \times 596 \times 0.46 + 0.308 \times 159 \times 0.39^{*} \\ &= 168\,\text{lm W}^{-1}\end{aligned} \qquad (2)$$

*The figure of 0.39 represents the average EQE of blue and green monochromatic QLEDs, considering that the green-blue $\beta$QLED contains nearly equal contributions from blue and green colors.

The practical efficacy reaches 47.5% of the ideal value and could be further improved with the implementation of more advanced light-extraction techniques, particularly focusing on enhancing efficiency for the orange-red and yellow-green color. It is important to acknowledge that, due to the detection noise floor at CEPREI, the driven voltage does not reach the actual sub-bandgap region, leading to an underestimation of efficacy. Based on measurements conducted in our laboratory, the real efficacy exceeds 181 lm W$^{-1}$ (above 50% of the ideal value), providing ample lumens for indoor lighting, equivalent to 4000 lumens from a $60 \times 120$ cm² flat panel troffer.

## Data availability

All data supporting the findings of this study are available within the paper and its Supplementary files. Any additional information related to the study is available from the corresponding author upon request.

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

## Acknowledgements

We acknowledge the support of all the technicians at Exciton Semiconductor Research Center of Zhejiang University. We also thank Prof. Dr. Xingliang Dai, Prof. Dr. Feng Liu, Prof. Dr. Hailong Fu, Dr. Zhiyuan Cao, Mr. Yunhao Lao and Mr. Yixuan Xu at Zhejiang University and Dr. Yunzhou Deng at University of Cambridge for their helpful discussions on device fabrication and characterizations. We are grateful to Dr. Yongtao Wang and Dr. Xinyu Wang at Zhejiang University for their valuable guidance in the EPR measurements. Additionally, we thank Dr. Yangjian Lin and Dr. Pei Sheng from Instrument and Service Center for Physical Science and Ms. Xin Li from Instrumentation and Service Center for Molecular Sciences at Westlake University for HAADF-STEM measurements and gas chromatography (GC) measurements, respectively. We also thank Dr. Pengfei Xu at China Electronic Product Reliability and Environmental Testing Research Institute (CEPREI) for assisting with certificated QLED measurement. We acknowledge financial support from the National Natural Science Foundation of China Nos. 22132005 (X.P.) and 62035013 (X.L.).

## Author contributions

X.P. supervised the entire project. X.L., Y.G., and X.P. conceived the idea. Y.Z. and X.L. designed the experiments, carried out the optical modeling and EPR measurements. Y.Z. and J.C. fabricated and characterized the high-performance QLED devices under the supervision of X.L. W.W. contributed to the ZnMgO and ZnO synthesis and characterizations. Z.S. performed the characterization of close-packed QD films under nanosecond laser excitation, supervised by X.L. H.W., and X.L. designed and built the setup for water-vapor treatment, electrical-pulse excitation, in situ and integrating-sphere-based *J-V-L* measurements. Y.G. assisted in the characterizations of all QDs and QLEDs. Y.Z., X.L., J.L., Z.H., Z.Y., H.Q., and X.P. participated in the data analysis. Y.Z., X.L., and X.P. prepared the manuscript with inputs from all the authors.

## Competing interests

The authors declare no competing interests.
