## [Transparent Peer Review file · Nature Communications]

In situ n-doped nanocrystalline electron-injection-layer for general-lighting quantum-dot LEDs

Corresponding Author: Professor Xiaogang Peng

Version 0:

Reviewer comments:

Reviewer #1

(Remarks to the Author)

This work is very interesting and innovative. Nano zinc oxide enables good performance as an electron transport layer for QLEDs, but its stability remains a significant concern in the industry. The author employed a water-vapor treatment method to achieve in-situ n-doping on the nanocrystalline electron injection layer, thereby enhancing electron transport and hole blocking capabilities, and achieving high brightness and efficiency of QLEDs under sub-bandgap driving voltage, potentially expanding the application of QLEDs in the field of white lighting. The referee support publication in Nature Communication, and there are some minor questions about this article:

1. A clear mechanism for the water-aluminum reduction reaction should be provided, along with details on the product changes before and after the reaction.
2. Figure 2 is based on green devices for discussion, while Figure 3 is based on red devices for discussion. Should there be consistency or clarification?
3. The color scheme in Figure 3 differs from previous figures, which may cause confusion;
4. In Figure 4abc, the summary of literature data has the vertical axis as the brightness of the device at 3V, and the meaning of this data is unclear;
5. In the devices tested with pulsed voltage, does the MgF2 insulating layer affect the water treatment effect?
6. SI Fig4d, the illustration is not clear enough and there is no legend;
7. SI Fig15 presents data on acid-encapsulated devices without discussing their performance, which seems unnecessary;
8. SI Fig17, the high energy peak of the device at 9V is not sufficient proof of ASE;

The author introduced the EPR test to characterize the transition of ZnMgO to n-type semiconductor, and conducted a large number of device performance tests and characterizations, with detailed demonstration processes. Overall, this work is worth publishing in Nature Communications.

Reviewer #2

(Remarks to the Author)

Please see the attachment

Reviewer #3

(Remarks to the Author)

In this work, the authors introduce reductive treatment to the EIL that effectively enhances the luminance and current density beyond the levels achieved with acid-containing resins. The newly-developed EIL paves the way for sub-bandgap-voltage-driven QD-blend LEDs for diffuse white-light sources. However, following questions needs to be discussed before reconsidering its publication.

1. Figure 1d indicates that the surface contact potential of ZnMgO is reverted after an oxygen treatment while current density

of the EOD using it is decreased more than its original current density with larger power factor. The author should suggest the reason of the discrepancy of the trend between surface contact potential and current density of the EOD.

2. Figure 2c shows the increased excitonic quenching by enhanced electron transfer to QDs. Also, Figure 2d shows the increased geometric capacitance after water-vapor treatment on the EIL by accumulation of electrons at the QD/ZnMgO interface. However, previous researches generally suggest that electron transfer to QDs and carrier accumulation at the QD/ZnMgO interface is the major origins of reduced EQE, while this research shows high EPE and IPE. The author should supplement the origin of enhanced efficiency to substantiate the claim.

3. The manuscript claims that the oxygen and water-vapor treatment cycles are “entirely reproducible,” but only two cycle of the oxygen and water-vapor treatment cycles are shown in Figure 2. It would be better to present additional cycles to claim the reproducibility.

4. In Supplementary Fig. 6, the authors claimed that the direct water-vapor treatment does not affect the HIL, HTL, or QD layers. However, if water treatment alone has no impact on the ETL, it seems more appropriate to first deposit Al and then perform water treatment, and then observe changes in the HIL, HTL, and QD layers.

5. For Figure 3 and Figure 4, optoelectrical characteristics of water-vapor treated QLED is emphasized. However, the explanation for why the electro-optical properties at V_p are crucial remains insufficiently addressed. Further explanation is required.

6. Figure 5b includes V_p value of 1.95 V. However, the prototype red β QLED includes three types of QDs with different bandgap energy, meaning that it can be inaccurate to present the bandgap energy of the mixed QDs with a simple V_p value.

7. Figure 2a shows the encapsulation using a resin. However, in line 151 of the main text, the encapsulation is referred to as “acid-free encapsulation”, while the term “resin” is typically used in the main text to refer to “acid-containing resin.” Therefore, it would be more appropriate to explicitly include the term “acid-free” in the figure caption.

8. In line 151, the phrase “ZnMgO/Al EIL/electrode” needs to be corrected to “ZnMgO EIL/Al electrode.”

Version 1:

Reviewer comments:

Reviewer #1

(Remarks to the Author)

The authors have addressed all my concerns, I recommend publication in Nature Communications.

I want to add some discussion regarding the referee 2's first comment.

In the paper mentioned by the referee, the QDs used are CdSe-CdS core-shell QDs, which have almost no electron confinement. Therefore, electron leakage is crazy, and less electron injection might be beneficial. However, for current widely used CdSe-ZnSe based QDs with better electron confinement, electron leakage is much less. Therefore, improving electron concentration benefits trap saturation.

I am listing some of the papers for the referee's potential interests:

1 Li, B. et al. Origin of the Efficiency Roll-off in Quantum Dot Light-Emitting Diodes: An Electrically Excited Transient Absorption Spectroscopy Study. *Nano Lett* 24, 10650-10655 (2024). <https://doi.org/10.1021/acs.nanolett.4c03024>

2 Bian, Y. Y. et al. Efficient green InP-based QD-LED by controlling electron injection and leakage. *Nature* 635, 854-859 (2024). <https://doi.org/10.1038/s41586-024-08197-z>

3 Gao, Y. et al. Minimizing heat generation in quantum dot light-emitting diodes by increasing quasi-Fermi-level splitting. *Nat Nanotechnol* 18, 1168-1174 (2023). <https://doi.org/10.1038/s41565-023-01441-z>

Reviewer #2

(Remarks to the Author)

All of my concerns have been addressed in the revisions. I think this article should be accepted by NC for publication.

Reviewer #3

(Remarks to the Author)

The authors have provided thorough responses to the previous review comments. Based on the significance and excellence of the content, I recommend the manuscript for publication.

Point-by-point Responses to the Reviewers

Responses to Reviewer 1:

Main comments:

This work is very interesting and innovative. Nano zinc oxide enables good performance as an electron transport layer for QLEDs, but its stability remains a significant concern in the industry. The author employed a water-vapor treatment method to achieve in-situ n-doping on the nanocrystalline electron injection layer, thereby enhancing electron transport and hole blocking capabilities, and achieving high brightness and efficiency of QLEDs under sub-bandgap driving voltage, potentially expanding the application of QLEDs in the field of white lighting. The referee support publication in Nature Communication, and there are some minor questions about this article:

Our revision and response: We thank the reviewer for recognizing the importance and scientific contribution of our work. Responses to all comments/suggestions pointed out by the reviewer are listed below.

Comment 1: *A clear mechanism for the water-aluminum reduction reaction should be provided, along with details on the product changes before and after the reaction.*

Our revision and response: We sincerely appreciate the reviewer's insightful comment regarding the reaction mechanism between the ZnMgO/Al interface and water vapor. We have provided additional experimental evidences to support the chemical mechanism during the revisions. The fundamental reactions of the n-type doping through the water-vapor treatment are illustrated in Figure 1f of the manuscript (figure to the right). We have reformulated these reactions in a more conventional manner in revised SI Fig. 2g and as follows:

Generation of hydrogen radicals:

Proton-coupled electron transfer:

Excess water-vapor treatment leading to H₂ formation:

Experimental evidences supporting these reactions include:

- (1) **Aluminum oxide generation** (revised SI Fig. 2h and figure below). The accumulation of excess O-species at the ZnMgO/Al interface, as revealed by the HAADF measurements in revised SI Fig. 2h (also figure below, left panel), provides evidence for the colocalization of Al, Zn, and O elements at the ZnMgO interface. Furthermore, XPS depth profile analysis of the water-vapor-treated devices (figure below, right panel) further confirms the localized formation of aluminum oxide at the ZnMgO/Al interface. Specifically, the XPS results reveal that, along the etching approaching the ZnMgO/Al interface, trivalent Al (Al³⁺) and oxygen species emerge almost simultaneously and sharply, while the Zn signals start to be gradually detectable and the metallic (Al⁰) signals drop drastically.

- (2) Excess electrons doped into the conduction band of ZnMgO nanocrystals. Upon the water-vapor treatment, UV-Vis (Fig. 1c, revised manuscript), EPR (Fig. 1b, revised manuscript), and surface contact potential (Fig. 1d, revised manuscript) measurements all confirm that excess electrons doped into the conduction band of ZnMgO. Specifically, UV-Vis measurements suggest that all ZnMgO nanocrystals are doped by the water-vapor treatment, with free electrons in the quantum-confined electron energy levels at the conduction band edge.
- (3) Hydrogen radicals (H^\cdot) and molecular hydrogen (H_2). EPR (revised SI Fig. 2f) measurements indicate generation of hydrogen radicals by the reaction of water vapor and metallic Al. Revised SI Fig. 2a-b reveals that, by a long (excessive) water-vapor treatment, formation of hydrogen bubbles is observed. Such a lengthy water-vapor treatment results in a detectable amount of molecular hydrogen by gas chromatography (revised SI Fig. 2d). Notably, hydrogen bubble formation is absent if aluminum is deposited directly on flat glass slide (revised SI Fig. 2c), suggesting that thermally evaporated aluminum atoms partially penetrate the nanoporous ZnMgO EIL, creating extensive surface areas that facilitate continuous Al-water reactions.

Hydrogen radicals refer to atomic hydrogen (H^\cdot), which is the known highly-reductive intermediate for formation of molecular hydrogen (H_2) by the reaction between an active metal and water. Albeit not as reductive, molecular hydrogen H_2 could also dope the ZnMgO nanocrystals according to their energy level alignment. In principle, the doping process with molecular hydrogen would result in the same products ($(ZnMgO)^{\cdot-} + H^+$) as those using atomic hydrogen. At this moment, we consider the above findings and references documented in literature on photochemical doping of oxide nanocrystals in solution provide substantial evidences for the proposed mechanism. Further detailed investigations to clarify the roles of hydrogen radicals vs. molecular hydrogen remain beyond the scope of this study.

In addition to the revisions in Figures mentioned above, several sentences are added on Page 6 of the revised manuscript: “Our experimental and analytical results suggest that the observed n-type doping efficiency stems from reducing hydrogen species produced during the Al-water reactions at the porous ZnMgO-Al interface. Specifically, Supplementary Fig. 2 provides direct evidence through detection of both hydrogen molecules (H_2) and hydrogen radicals (H^\cdot), substantiating this proposed mechanism. Although molecular hydrogen with moderate reductivity could transfer their electrons to the ZnMgO nanocrystals (Supplementary Fig. 3) followed by proton formation, its delayed detection relative to the onset of current and luminance enhancement (Supplementary Fig. 2e) and weaker reducing power compared to atomic hydrogen suggest H_2 plays a secondary role. Thus, we primarily attribute the doping mechanism to hydrogen radical-mediated electron transfer, though complete elucidation of their interfacial stabilization mechanisms awaits future study.”

Comment 2: Figure 2 is based on green devices for discussion, while Figure 3 is based on red devices for discussion. Should there be consistency or clarification?

Our revision and response: Thanks for noting us about this potential issue of consistency. The observed effects of n-type doping on the EIL in QLED devices (Figure 2) demonstrate consistent behavior across different emission colors, as the underlying mechanisms for performance enhancement remain fundamentally identical. Specifically, the reductive treatment effectively addresses the inherent limitations of pristine ZnMgO EIL, notably its poor electron conductivity and hole blocking characteristics, which are universal across the blue, green and red-emitting QLEDs. While Figure 2 primarily presents data from green-emitting devices to illustrate these effects, we have selected red-emitting QLEDs as a representative example in Figure 3 to demonstrate the broad applicability of our findings. We provide performance metrics for multi-color QLEDs in SI Figs. 11–14.

To clarify this point, we have added the following statement on Page 14 of the revised manuscript before discussing performance of all devices: “Given the inherent limitations of pristine ZnMgO EILs, specifically their inadequate electron conductivity and subpar hole-blocking capabilities, we anticipate a universal enhancement trend in water-vapor treated QLEDs spanning all emission colors (see detail below).”

Comment 3: *The color scheme in Figure 3 differs from previous figures, which may cause confusion.*

Our revision and response: We thank the reviewer for bringing this important observation to our attention. In response to the comment regarding the color scheme inconsistency, we have carefully revised Figures 2-3 to maintain visual consistency throughout the manuscript.

Comment 4: *In Figure 4abc, the summary of literature data has the vertical axis as the brightness of the device at 3V, and the meaning of this data is unclear.*

Our revision and response: We appreciate the opportunity to clarify the rationale for benchmarking device performance at 3V. Our justification is primarily based on general lighting applications.

While our QLEDs achieve maximum luminance at relatively high voltages, such high-voltage operation inherently compromises external power efficiency (EPE). As derived from the relationship $EPE = (LEE \times IQE) \times V_p/V = EQE \times V_p/V$, lowering the operating voltage (V) directly enhances EPE. This principle aligns with global efforts to improve energy efficiency in general lighting industries. Notably, the U.S. Department of Energy 2022 Solid-State Lighting R&D Report (Pattison, M. *et al.* 2022 DOE SSL R&D Opportunities) explicitly prioritizes voltage reduction, with 3V designated as a 2035 performance as well as the final target (table to the bottom, adopted from Table 4.3, 2022 DOE SSL report). By adopting this standardized voltage for comparison (as commonly practiced in QLED literature—see references in Fig. 4a-c), our work contextualizes advancements within both academic and industrial roadmaps.

Response Tab. 1, quoted from reference

[REDACTED]

To clarify this issue, two sentences are added on Page 15 in the revised manuscript: “The U.S. Department of Energy (DOE) 2022 Solid-State Lighting R&D Report identifies operational voltage minimization as a pivotal strategy for achieving energy-efficient general lighting, setting 2035 performance targets at luminance of 8000 cd/m² at 3.0 V bias. Though challenging, achieving operating voltages near or below V_p would be ideal. The reductive treatments introduced here are readily extended to QLEDs across the visible spectrum, enabling low-voltage operation with high efficiency. Experimental results reveal 2.6-12.3 times luminance enhancement at V_p compared to conventional devices (Fig. 4a–c, Supplementary Table 1 and Figs. 12-15). As a result, the luminance level at 3 V substantially exceeds the DOE targets across the visible spectrum (Fig. 4a-c).”

Comment 5: *In the devices tested with pulsed voltage, does the MgF₂ insulating layer affect the water treatment effect?*

Our revision and response: We thank the reviewer for raising this important point regarding the potential impact of the MgF₂ insulating layer on the water-vapor treatment process. Below, we clarify the role and compatibility of this layer:

The 50-nm MgF₂ insulating layer is *selectively* deposited via shadow-mask thermal evaporation between the ZnMgO EIL and the patterned Al cathode. This localized deposition ensures that MgF₂ *only physically blocks current flow in its covered regions*, while the remaining active areas (uncoated by MgF₂) remain fully accessible to water-vapor treatment. In principle, any insulating material inserted at any interface in the multi-layer structure (e.g., ITO/HTL, HTL/QD, QD/ZnMgO, etc.) achieves the same current confinement without affecting the water-vapor treatment effect.

Comment 6: *SI Fig4d, the illustration is not clear enough and there is no legend.*

Our revision and response: We apologize for the oversight regarding the clarity of SI Fig. 4d. The Figure becomes SI Fig. 5d in the revised manuscript, and it is now presented with a clear legend (figure to the right).

Comment 7: *SI Fig15 presents data on acid-encapsulated devices without discussing their performance, which seems unnecessary.*

Our revision and response: We appreciate the reviewer’s valuable comment regarding the presentation of acid-encapsulated device data. We agree that the inclusion of these data without corresponding performance analysis may distract from the main focus of our study. Therefore, we have removed the acid-encapsulation related data (SI Fig. 16, figure to the right) from the revised manuscript to maintain clarity and focus on the core findings of our research.

Comment 8: *SI Fig17, the high energy peak of the device at 9V is not sufficient proof of ASE.*

Our revision and response: We thank the reviewer for pointing out this issue. We agree that the emergence of high energy peak in EL does not mean the realization of electrically pumped ASE. To avoid misleading, we re-write the related part in the revised manuscript (Page 16): “For instance, in red-emitting QLEDs, a distinct high-energy emission peak emerges when the radiance exceeds $\sim 20,136$ W/sr·m² under 9 V bias, exhibiting spectral characteristics similar to those induced by pulsed laser excitation (Supplementary Fig. 18). This phenomenon can be attributed to the recombination process involving p-shell electrons, suggesting the occurrence of population inversion in the s-shell energy states.” In addition, the optically pumped ASE data previously included in Supplementary Fig. 17 has been removed, as this falls outside the scope of our studies on electrically driven devices. The revised figure now focuses solely on electroluminescence spectral evolution under varying bias conditions.

Comment 9: *The author introduced the EPR test to characterize the transition of ZnMgO to n-type semiconductor, and conducted a large number of device performance tests and characterizations, with detailed demonstration processes. Overall, this work is worth publishing in Nature Communications.*

Our revision and response: Thanks for the encouragement. No action needed here.

Responses to Reviewer 2:

Main comments:

This manuscript presents an effective and simple method to achieve heavily n-doped magnesium-doped ZnO (ZnMgO) via in-situ reductive treatment of water-vapor with Al electrode, which can be used as electron-injection layer (EIL) for QLED or other optoelectronic devices. The resulting QLEDs exhibit optimal efficiency and extraordinarily-high brightness. This breakthrough work not only effectively eliminates the long-standing issue of device positive aging induced by acid-containing encapsulation, but also makes QLED lighting possible. I recommend it be published after minor revisions. Some issues should be addressed:

Our revision and response: We thank the reviewer for recognizing the importance and scientific contribution of our work. Responses to all comments/suggestions pointed out by the reviewer are listed below.

Comment 1: *It is currently widely believed that there is excessive electron injection and insufficient hole injection in devices, and many studies have shown that improving hole injection or suppressing electron injection can enhance device performance (Nature, 2014, 515, 96–99; Sci. Adv. 10, 2024, eado0614; Adv. Mater. 2023, 2303950). However, in studies on positive aging, the general conclusion is that positive aging originates from the increased conductivity of the electron transport layer (ETL), which enhances electron transport efficiency. This study focuses on improving the conductivity of ZnMgO, and the resulting devices exhibit excellent performance (optimal efficiency and extraordinarily-high brightness), and successfully eliminates the positive aging issue. How could this seemingly contradictory phenomenon be explained?*

Our revision and response: We deeply appreciate the reviewer's insightful comment regarding this crucial aspect of our work, which indeed represents one of the most significant contributions of our study. We apologize for not describing this important discovery clearly.

Yes, studies on QLEDs encapsulated with the acidic resins in this work (Revised SI Fig. 8 and 11-14) and in the literature all point out that positive aging originates from the improved electron injection and conductivity of the ZnMgO ETL. The water-vapor treatment to the devices with the Al electrodes without any resin encapsulation actually completes necessary improvements of the ETL efficiently and controllably. After encapsulation with the acid-free resins, fully optimized (in terms of their ETL) devices would immediately show outstanding performance yet eliminate both positive aging and destructive aging observed for the devices encapsulated with the acidic resins in long term storage/operation.

Enhancing the hole injection is important for either conventional devices with the acidic resin or our devices with the water-vapor treatment. In the water-vapor treated devices, this is achieved by Coulomb attraction of the injected electron in a QD as described in previous reports (see figure to the right from Deng *et al.*, *Nat. Commun.* **11**, 2309, 2020). Basically, acceleration of the electron injection would greatly promote injection of the holes (even ~2 times faster than the initial electron injection). This is so because the injected electron is spatially-confined within a QD and would thus elevate both hole and electron energy levels for efficient hole injection and hindered electron

[REDACTED]

injection into the QD.

To further enhance this point, one sentence is added on Page 13-14 in the revised manuscript: “In principle, the reductive treatment efficiently and controllably maximizes all necessary enhancements on the electron injection and transport of the ZnMgO ETL prior to completion of the device fabrication by the acid-free encapsulation, which also pre-excludes all negative aging effects caused by the acidic resins in the relatively long term (Supplementary Fig. 11-14).”

By the way, thanks for bringing a new reference (*Adv. Mater.* 2023, 2303950) into our attention, which is now cited as **Ref. 26** in related places in the revised manuscript. In principle, further improvement of the new devices with the water-vapor treatment might come from greatly accelerating the hole injection. Furthermore, this is an excellent reference to justify significance of our work as well as to compensate our efforts on improving hole injection.

Comment 2: *ZnO has stronger n-type characteristics and better electron transport properties than ZnMgO. Why did the authors choose not to use ZnO in this work? What was the rationale behind this choice?*

Our revision and response: Thanks for reminding us this issue. This decision was based on both fundamental material considerations and practical device requirements, supported by existing literature and our experimental findings. While bulk ZnO is indeed an n-type semiconductor with favorable electron affinity for electron transport/injection, ZnO nanocrystals exhibit fundamentally different electronic properties. As demonstrated by Liu *et al.* (*Phys. Rev. Lett.* **98**, 186804, 2007) and Schrauben *et al.* (*Science* **336**, 1298-1301, 2012), pristine ZnO nanocrystals lack free carriers, as evidenced by the absence of conduction band electron signals in UV-Vis, IR and EPR measurements. Although photochemical n-doping of ZnO nanoparticles in solution has been reported (*Phys. Rev. Lett.* **98**, 186804, 2007), electron injection via ZnO ETL (optimized with the acidic resin) has been demonstrated to be suitable for red-emitting devices but not for the green- and blue-emitting ones. Consequently, ZnMgO, with its tunable and smaller electron affinity, has been widely adopted as the preferred ETL for the QLEDs, especially the green- and blue-emitting ones. Given our focus on developing QLEDs for general lighting applications, which requires high efficacy at ultra-brightness across the entire visible spectrum, ZnMgO represents the most rational choice.

It should be noted that the water-vapor treatment is found to be effective to the devices with the ZnO ETL as well. Because of the limited length, this was only mentioned in a few words in the original submission. One sentence is added on Page 16 in the revised manuscript: “While our methodology development primarily centers on the ZnMgO systems, the demonstrated universality is evidenced by enhanced performance in ZnO-based QLEDs (Supplementary Fig. 20).”

Comment 3: *EPR is a powerful tool for analyzing the magnetic properties, defects, and free charge carriers in materials by detecting the spin characteristics of unpaired electrons. However, it is difficult to distinguish by EPR signal whether these unpaired electrons originate from defects/impurities, or from conduction band states. It is suggested to complement this with temperature-dependent conductivity or Hall effect experiments to investigate n-type characteristics of ZnMgO or ZnO.*

Our revision and response: We thank the reviewer for this important methodological consideration. While EPR spectroscopy indeed detects all unpaired electrons regardless of their origin, we have conducted multiple

complementary characterizations to confirm the conduction band nature of the introduced charge carriers.

Thanks to the photochemical doping studies on colloidal ZnO (and other oxide) nanocrystals in solution, UV-Vis, IR, and EPR measurements are confirmed to be sufficient to assure the location of unpaired electrons in the ZnMgO (or ZnO) nanocrystalline ETL (*Phys. Rev. Lett.* **98**, 186804, 2007, *Science* **336**, 1298-1301, 2012). While EPR measurements are only sensitive to chemical species with unpaired electrons, the observed g-factor (1.970) from the water-vapor-treated ZnMgO/Al closely matches delocalized electrons in the quantum-confined electron energy levels near the conduction band edge of ZnO nanoparticles (1.969). More importantly, the correlated UV-Vis and IR measurements are not only sensitive to doped electrons, which unambiguously confirm nearly each ZnMgO (or ZnO) nanocrystal in the ETL upon the water-vapor treatment is filled with at least one electron in the quantum-confined electron energy level(s) near the conduction band edge.

Unfortunately, Hall effect and temperature-dependent conductivity measurements (Figure to the right) failed to yield informative conclusions. Specifically, the detection limit of conventional Hall techniques ($0.1 \text{ cm}^2/(\text{V}\cdot\text{s})$) is much higher than the electron mobility of our nanocrystalline ZnMgO ETL. Temperature dependent conductivity alone is known to be a complex of several effects, mostly about the thermal activation of electron hopping as documented in literature (Zabet-Khosousi, et al. *Chemical Reviews*, **108**, 10, (2008)). As shown in Figure to the right, we did observe significant temperature dependence of the electron-only-device (EOD) after the water-vapor treatment, but it is hard to draw clear conclusion. The results reveal a tenfold increase in Ohmic-region resistivity at 120 K compared to room temperature, tentatively attributed to reduced free carrier concentration and diminished thermal-assisted hopping transport rate. In the high-current regime, the J-V slope increases from 1 (Ohmic) to 1.59 with cooling, deviating from both Child's law ($J \propto V^2$) and trap-limited space-charge-limited conduction models.

Albeit indirectly, two key electronic measurements strongly support the reductive doping of the ZnMgO ETL, i.e., the significantly reduced surface contact potential demonstrating Fermi level upshift (Fig. 1d) and much enhanced electron conductivity by 3-4 orders of magnitude (Fig. 1e).

One sentence is added on Page 12 in the revised manuscript to further elaborate this issue: “These findings demonstrate that strategically incorporating electrons into the quantum-confined states at the conduction band edge of ZnMgO nanocrystals significantly enhances two critical processes: (1) electron transport within the EIL and (2) interfacial electron transfer from ZnMgO to QDs under forward-bias.”

Comment 4: After water-vapor treatment of ZnMgO/Al, in what form does the Al electrode exist after losing electrons? Does it exist as AlO or Al₂O₃? Will this affect the energy levels and conductivity of ZnMgO? For QLED devices, the energy level configurations of the electrode, HIL and quantum dot light-emitting layer is another key factor regulating electron injection/transport. It is recommended to provide the energy level arrangement of QD, ZnMgO before and after treatment and Al electrode (including Ti, Au and Ag) and add the corresponding discussion.

Our revision and response: We sincerely thank the reviewer for raising these critical points regarding the chemical and electronic changes of the ZnMgO EIL and Al electrode induced by the water-vapor treatment. Below, we provide

a detailed response addressing the concerns about interfacial chemistry, energy level alignment, and their implications for device performance:

- (1) Existence of Al oxide (Al^{3+}) at the interface between the Al electrode and the ZnMgO ETL. An extremely thin aluminum oxide layer is detected at the ZnMgO-Al interface by HAADF in revised SI Fig. 2h and figure (left panel). At this point, we can only identify trivalent Al^{3+} by the XPS measurements (see figure below, right panel) and cannot tell the exact structure of the aluminum oxide. In either case, we can estimate the thickness of the non-continuous aluminum oxide layer would be < 1 nm in thickness by the electrons transferred into the ZnMgO nanocrystals, especially taking into account the roughness of the Al electrode deposited on the nanoporous ZnMgO ETL. We would consider such a thin and non-continuous aluminum oxide layer should not affect conductivity of the ZnMgO layer, which is in good agreement with the much-improved electron transport and electron injection of the ETL by the water-vapor treatment.

- (2) Energy level alignments for electron injection. The flat-band energy level alignments of the QD/ZnMgO/Al system in the revised SI Fig. 3 and Figure to the right (left panel) is consistent with literature reports. Although the Fermi level of the ZnMgO ETL shifts by ~ 0.7 eV after the water-vapor treatment with the Al electrode (Fig. 1d in the revised manuscript), it remains deeper than that of Al, Ti, and Ag but higher than Au. The alignments are found to be consistent with the observed trends in the current density of the devices with Al, Ti, Ag and Au electrodes.

Besides addition of the revised SI Fig. 3 partially discussed above, description about the AlO_x is provided in figure caption of revised SI fig. 2: “Since all free electrons in the ZnMgO conduction band originates atomic Al, the resulting AlO_x layer formed at the Al-ZnMgO interface through water-vapor treatment is estimated to be < 1 nm and discontinuous, which should not significantly impede the electron conductivity of the ETL.” As for the energy

alignments and electron injection/transport, one sentence is added on Page 9-10 in the revised manuscript: “After the water-vapor treatment with the Al electrode, the electron injection and transport of the ZnMgO EIL should be greatly improved according to the energy level alignments in Supplementary Fig. 3.”

One additional plot is added in revised SI Fig. 3 (see Figure above, right panel) and one sentence is added on Page 16-17 in the revised manuscript: “According to the electrode potential in Supplementary Fig. 3, reduction potentials of the Ag/Ag⁺ and Au/Au⁺ (or Au/Au³⁺) redox pairs are too low to initiate the hydrogen reduction reaction.”

Comment 5: *In Figure 2g, the current density increases sharply above ~0.6 V but decreases below this voltage during the initial water-vapor treatment. The authors attribute that the in situ reductive treatment with the Al electrode greatly improves both hole blockage and electron conductivity of the ZnMgO EIL, which are respectively associated with deep recombination centers and free-electron concentrations within the ZnMgO EIL. However, for QLED devices, in the low voltage region (~0.7 eV) the current is mainly contributed by electrons and hole filling, and the holes have not yet been injected into the QD layer. Therefore, there seems to be no effect on hole transport at this time. This issue should be addressed further.*

Our revision and response: We sincerely appreciate the reviewer’s insightful observation regarding the low-voltage region behavior. This phenomenon can be explained through the following comprehensive analysis:

- (1) **Defect-Mediated Hole Leakage Mechanism.** While we agree that holes cannot directly transfer from the quantum dot valence band to ZnMgO at low bias voltages due to insufficient hole population in QDs, the quasi-monolayer morphology of the QD layer allows direct contact between ZnMgO and the HTL. As demonstrated in previous studies (Chen, Z. *et al. Nano Res.* **14**, 1 (2021) and Luo, H. *et al. ACS Nano* **13**, 7 (2019)), defects in pristine ZnMgO can serve as efficient hole leakage pathways, bypassing the QD layer entirely. Hole leakage also occurs at higher voltages, but would be drowned out by recombination current at QD layer.
- (2) **Experimental Evidence Supporting Defect Passivation:** The effectiveness of the water-vapor treatment in addressing this issue is supported by multiple experimental observations. Significant reduction in visible photoluminescence from ZnMgO nanocrystals indicates defect passivation (SI Fig. 10a). Improved open-circuit voltage in photovoltaic mode operation (SI Fig. 10b) also implies removal defects by the water-vapor treatment. Evidently, suppression of the leakage current in the low-voltage region post-treatment (Fig. 2g and SI Fig. 9b, bottom panel) is consistent with all these measurements.
- (3) **Comprehensive Impact of Reductive Treatment:** The water-vapor treatment simultaneously enhances electron conductivity through n-doping and reduces hole leakage by passivating deep-level defects. This dual mechanism explains the observed current density behavior and highlights the importance of defect passivation in optimizing QLED performance.

On Page 11 in the revised manuscript, one sentence is added to clarify the issue: “The presence of only ~1 monolayer of QDs between the HTL and EIL creates a potential pathway for direct hole leakage to the EIL trap states below the turn-on voltage³⁷⁻³⁹.”

Comment 6: *For QLED, the ultimate factor affecting its performance is the injection of electrons and holes into the light-emitting layer of quantum dots. The authors significantly improved the electrical conductivity of ZnMgO HIL through a simple in-situ reductive treatment, resulting in unprecedented device performance. In order to be able to apply this in-situ reductive doping method widely, the mechanism of n-type doping and the resulting efficient injection*

of charge into the quantum dot layer should be further discussed in detail.

Our revision and response: We very much appreciate the reviewer's insightful comments regarding the mechanism of n-type doping and its critical role in charge injection optimization. To address this fundamental aspect, we decide to add a comprehensive discussion of the doping mechanism and its consequences for device performance, supported by multiple lines of experimental evidences.

The doping mechanism is illustrated as a set of reactions provided in the revised SI Fig. 2:

Experimental evidences supporting this set of reactions and the doping mechanism include:

- (1) Aluminum oxide generation. This has been discussed in detail above in the Response to Comment 4. In short, an ultra-thin and non-continuous AlO_x layer is identified at the ZnMgO/Al interface.
- (2) Excess electrons doped into the conduction band of ZnMgO nanocrystals. Upon the water-vapor treatment, UV-Vis (Fig. 1c, revised manuscript) and EPR (Fig. 1b, revised manuscript) all confirm that free electrons are doped into the conduction band of ZnMgO nanocrystals in the ETL. Specifically, UV-Vis measurements suggest that all ZnMgO nanocrystals are doped by the water-vapor treatment, with about 1~2 free electrons in the quantum confined electron energy levels at the conduction band edge.
- (3) Hydrogen radicals (H^\cdot) and molecular hydrogen (H_2). EPR (revised SI Fig. 2) measurements indicate generation of hydrogen radicals by the reaction of water vapor and metallic Al. Revised SI Fig. 2 also reveals that, by a long (excessive) water-vapor treatment, formation of hydrogen bubbles are observed. Such a lengthy water-vapor treatment results in a detectable amount of molecular hydrogen by gas chromatography (revised SI Fig. 2d). Notably, hydrogen bubble formation is absent if aluminum is deposited directly on glass slide (revised SI Fig. 2c), suggesting that thermally evaporated aluminum atoms partially penetrate the porous ZnMgO EIL, creating extensive surface areas that facilitate the continuous Al-water reactions.

The mechanism on the efficient electron injection into the QD layer. Mechanism on the efficient charge injection enabled by the water-vapor treatment is further elaborated by adding the revised SI Fig. 3, which is prepared according to this reviewer's suggestion and also demonstrated above as the second figure in Response to Comment 4. Basically, the Femi level of the ZnMgO ETL increases for ~0.7 eV by the water-vapor treatment with the Al electrodes (Fig. 1d), and the electron transport/injection is improved by 3~4 orders of magnitude (Fig. 1e and SI Fig. 5). Consequently, the hole injection is greatly enhanced by Coulomb attraction of the injected electron in a QD as described in the previous reports. However, further improvement of the new devices might come from accelerating the hole injection.

Comment 7: *This work has demonstrated ultra-bright and energy-efficient QLEDs across the visible spectrum. In Figure 4, all of the three red, green and blue QLED devices show ultra-bright and efficient at very high driving voltages. What is the possible mechanism?*

Our revision and response: Sorry for not making it clear. The QLEDs with the water-vapor treatment are not intended to operate at very high driving voltages. In fact, all devices in Fig. 4 reach their maximum brightness at ~10 V. Because of their extremely high current/brightness at a medium bias, the heat dissipation becomes a serious issue for a device at very high driving voltages with generic structure/materials. In a way, the excellent efficiency at ultra-

high brightness across the visible spectrum is realized by the largely reduced driving voltages at a given brightness.

In Response to Comment 1, we quoted the sequential injection mechanism for QLEDs (Deng *et al.*, *Nat. Commun.* **11**, 2309, 2020). Initial injection of an electron into a neutral QD creates Coulomb potential wells that accelerate subsequent hole injection and impedes additional electron injection. Results in literature have repeatedly suggested that the sequential injection model works for different color QLEDs. Here, for the QLEDs prepared by the water-vapor treatment, the same mechanism is reconfirmed across the visible spectrum. As shown in Fig. 5a in the revised manuscript (also figure to the left), the injection voltage at a pretty high brightness (current density 1 mA/cm²) across the visible

spectrum is all ~ 0.2 V below the corresponding optical bandgap.

Two sentences are added to point out the low driving voltage with record-high brightness on Page 15-16 in the revised manuscript: “Our QLEDs across the visible spectrum reaches unpresented maximum luminance (Fig. 4d, red: 1,250,000, green: 3,102,000, and blue: 207,000 cd/m²), limited by electron leakage²⁶ or thermal quenching effects. Notably, the driving voltages at the record-high brightness for all colors are quite low (only ~ 10 V).” Along with citing the new reference suggested by the reviewer (*Adv. Mater.* 2023, 2303950), two sentences is added in the revised manuscript to further address the charge injection, on Page 16: “We expect that the performance of our device would be further boosted via enhanced hole injection and blocked electron leakage²⁶.” and on Page 18: “It is anticipated that, by further improving the hole injection²⁶, the driving voltage at the given luminance range (600-20,000 cd/m²) could be reduced for achieving IPE significantly over unity.”

Responses to Reviewer 3:

Main comments:

In this work, the authors introduce reductive treatment to the EIL that effectively enhances the luminance and current density beyond the levels achieved with acid-containing resins. The newly-developed EIL paves the way for sub-bandgap-voltage-driven QD-blend LEDs for diffuse white-light sources. However, following questions needs to be discussed before reconsidering its publication.

Our revision and response: We thank the reviewer, no action needed here.

Comment 1: Figure 1d indicates that the surface contact potential of ZnMgO is reverted after an oxygen treatment while current density of the EOD using it is decreased more than its original current density with larger power factor. The author should suggest the reason of the discrepancy of the trend between surface contact potential and current density of the EOD.

Our revision and response: Sorry for not making it clear. The discrepancy can be elaborated by considering the

different material states and their corresponding electronic properties. In Figure 1 (also Figure to the right), EPR in *b* and UV-Vis in *c* both confirm that the oxygen treated sample after the water-vapor treatment (quoted as “/H₂O/O₂” in *b* and *e* or “ZnMgO/Al/H₂O/O₂” in *c*) is in the same electronic state of the “pristine” ZnMgO, instead of the ZnMgO film with the Al deposition (quoted as “ZnMgO/Al” in *b*, *c*, and *e*). Figure to the right (*d*) illustrates that, upon deposition of the Al electrode onto the “pristine” ZnMgO film, the surface contact potential is already reduced somewhat, consistent with the partial doping state by the Al deposition in *b* and *c*. While the water-vapor treatment further reduces the surface contact potential, the subsequent oxygen treatment actually reverts all effects of both Al deposition and following water-vapor treatment on the surface contact potential in *d*, consistent with the EPR and UV-Vis measurements. However, without a top electrode, it would be impossible to construct any EOD, and thus in *e*, there are only J-V measurements for “ZnMgO/Al” and “/H₂O/O₂”, in addition to two curves ended with the water-vapor treatment.

To clarify this issue, one sentence on Page 7-8 in the revised manuscript is added: “It should be noted that, in addition to two overlapping curves both ended with the water-vapor treatment, the J-V measurements in Fig. 1e only include the ZnMgO/Al (the ZnMgO layer with the Al electrode) and /H₂O/O₂ (the sample after subsequent water-vapor and oxygen treatments).”

Comment 2: Figure 2c shows the increased excitonic quenching by enhanced electron transfer to QDs. Also, Figure 2d shows the increased geometric capacitance after water-vapor treatment on the EIL by accumulation of electrons

at the QD/ZnMgO interface. However, previous researches generally suggest that electron transfer to QDs and carrier accumulation at the QD/ZnMgO interface is the major origins of reduced EQE, while this research shows high EPE and IPE. The author should supplement the origin of enhanced efficiency to substantiate the claim.

Our revision and response: We sincerely appreciate the reviewer's insightful observation regarding the contradiction between enhanced excitonic quenching and improved device efficiency. This phenomenon can be explained through a comprehensive analysis of the interfacial charge dynamics and operational mechanisms in our optimized QLED architecture.

The steady-state PL quenching and transient PL lifetime reduction are measured on the **unbiased** devices (Fig. 2c and inset), which are primarily originated from efficient electron transfer from the QDs to the ZnMgO layer, rather than reverse charge injection. Under the unbiased condition, photogenerated electrons in the QD layer naturally migrate toward the cathode (ZnMgO/Al electrode) while holes drift toward the anode (HIL/HTL), mimicking photovoltaic behavior in solar cells with an increased open-circuit voltage upon the water-vapor treatment (SI Fig. 9b).

The operational paradigm of QLEDs under **forward bias** fundamentally alters the charge dynamics compared to the unbiased conditions. Basically, free electrons are readily driven to the ZnMgO nanocrystals adjacent to the QDs in QLEDs (Fig. 2d), injected into the QDs, and override the build-in potential. Upon the water-vapor treatment, the electron transport/injection becomes greatly improved and strongly promotes this series of effects.

To clarify experimental conditions, we have amended the caption of Fig. 2c to explicitly state: “All photoluminescence measurements were performed on the unbiased devices under open-circuit conditions.”

Comment 3: *The manuscript claims that the oxygen and water-vapor treatment cycles are “entirely reproducible,” but only two cycle of the oxygen and water-vapor treatment cycles are shown in Figure 2. It would be better to present additional cycles to claim the reproducibility.*

Our revision and response: We sincerely appreciate the reviewer's valuable comment regarding the reproducibility demonstration. We fully agree that the initial claim of ‘entirely reproducible’ was premature based on only two treatment cycles. The oxygen treatment (including its reversibility) is not at the center of this work, and we thus have revised the manuscript accordingly on Page 11: “Furthermore, an additional water-vapor treatment can bring the devices back to the state before the oxygen treatment...”

Comment 4: *In Supplementary Fig. 6, the authors claimed that the direct water-vapor treatment does not affect the HIL, HTL, or QD layers. However, if water treatment alone has no impact on the ETL, it seems more appropriate to first deposit Al and then perform water treatment, and then observe changes in the HIL, HTL, and QD layers.*

Our revision and response: This is an important question but we thought it is likely out of the scope of the current work. In any case, the following experiments are carried out to address this issue from two different perspectives.

(1) Effects on HTL, HIL, and QDs of the water-vapor treatment of the devices without the ZnMgO EIL. We fabricated devices following the standard recipe while omitting the ZnMgO layer. As expected, direct contact of the Al electrode onto the QD layer completely quenches their photoluminescence and the subsequent water-vapor

treatment does not recover the emission (Photos in top row, Figure to the top). The Al electrodes of both types of the devices—with or without the water-vapor treatment—are removed, and fully functional devices with the ZnMgO EIL are reconstructed without the water-vapor treatment. Measurements in Figure to the top (a, b, and c) show nearly identical performance. Given the destruction of QDs by the Al deposition, we think these results can only tell there is no additional effects to HTL, HIL, and QDs by the water-vapor treatment with the Al electrode.

- (2) Effects on HTL, HIL, and QDs of the water-vapor treatment of the devices with the ZnMgO EIL. The top ZnMgO/Al layers of two original devices with all layers—one with and the other without the water-vapor treatment—are removed, and deposition of fresh ZnMgO and Al layers without the water-vapor treatment completes the reconstruction. Results in Figure to the top (d, e, and f) confirm that the two reconstructed devices show nearly identical performance, indicating the water-vapor treatment of a fully functional device with all layer also does not affect the HTL, HIL, and QDs.

We note that the peel-off-rebuild process itself needs some further optimization to be completely reproducible. Considering the manuscript is already long, we would not include the information above in the revised manuscript. With the existing experimental results, we believe it is safe to claim that “direct water-vapor treatment on the HIL, HTL, and QD layers does not affect their optical or device optoelectronic properties”. As for effects of the reductive doping process on these layers, we would leave it out for the current publication.

Comment 5: For Figure 3 and Figure 4, optoelectrical characteristics of water-vapor treated QLED is emphasized. However, the explanation for why the electro-optical properties at V_p are crucial remains insufficiently addressed. Further explanation is required.

Our revision and response: We appreciate the opportunity to clarify the rationale for benchmarking device

performance at V_p and 3V. As derived from the relationship $EPE = (LEE \times IQE) \times V_p/V = EQE \times V_p/V$, lowering the operating voltage (V) directly enhances EPE. This principle aligns with global efforts to improve energy efficiency in general lighting industry. The U.S. Department of Energy’s 2022 Solid-State Lighting R&D Report (Pattison, M. *et al.* 2022 DOE SSL R&D Opportunities) explicitly prioritizes voltage reduction, with 3.7, 3.3, and 3V designated as a performance target for year 2021, 2025, and 2035, respectively (table below, adopted from Table 4.3, 2022 DOE SSL report). Ideally, if devices with their operational voltage at V_p (or even lower than V_p) could be sufficiently bright (~8000 nit), that would be the best. By adopting these standardized voltages for comparison (as commonly practiced in QLED literature—see references in Fig. 4a-c), our work contextualizes advancements within both academic and industrial roadmaps.

Response Tab. 1, quoted from reference

[REDACTED]

To clarify this issue, several sentences are added on Page 15 in the revised manuscript: “The U.S. Department of Energy (DOE) 2022 Solid-State Lighting R&D Report identifies operational voltage minimization as a pivotal strategy for achieving energy-efficient general lighting, setting 2035 performance targets at luminance of 8000 cd/m² at 3.0 V bias. Though challenging, achieving operating voltages near or below V_p would be ideal. The reductive treatments introduced here are readily extended to QLEDs across the visible spectrum, enabling low-voltage operation with high efficiency. Experimental results reveal 2.6-12.3 times luminance enhancement at V_p compared to conventional devices (Fig. 4a–c, Supplementary Table 1 and Figs. 12-15). As a result, the luminance level at 3 V substantially exceeds the DOE targets across the visible spectrum (Fig. 4a-c).”

Comment 6: *Figure 5b includes V_p value of 1.95 V. However, the prototype red β QLED includes three types of QDs with different bandgap energy, meaning that it can be inaccurate to present the bandgap energy of the mixed QDs with a simple V_p value.*

Our revision and response: We sincerely appreciate the reviewer’s insightful comment regarding the representation of V_p in our mixed QD system. The red β QLED indeed incorporates three distinct QD populations with emission peaks at 605 nm (2.05 eV), 623 nm (1.99 eV), and 644 nm (1.93 eV), creating a complex energy landscape. To accurately characterize this system, we have employed spectral intensity-weighted integration of the photoluminescence profile to determine the ‘average photon energy’, which provides a theoretically justified basis for the reported V_p value of 1.95 V. This approach accounts for the asymmetric energy distribution of the emission spectrum and the relative contribution of each QD population to the overall device performance.

To ensure clarity and prevent potential misinterpretation, we have revised the Figure 5 caption to explicitly state: “ V_p represents the average photon energy calculated through spectral intensity-weighted integration of the electroluminescence profile, accounting for the contributions from all QD components in a β QLED.”

Comment 7: *Figure 2a shows the encapsulation using a resin. However, in line 151 of the main text, the*

encapsulation is referred to as “acid-free encapsulation”, while the term “resin” is typically used in the main text to refer to “acid-containing resin.” Therefore, it would be more appropriate to explicitly include the term “acid-free” in the figure caption.

Our revision and response: We thank the reviewer for the kind suggestion and ‘acid-free resin’ notation is added to Fig. 2a in the revised manuscript.

Comment 8: *In line 151, the phrase “ZnMgO/Al EIL/electrode” needs to be corrected to “ZnMgO EIL/Al electrode.”*

Our revision and response: Sorry for this careless mistake and it is corrected in the revised manuscript.

This manuscript presents an effective and simple method to achieve heavily n-doped magnesium-doped ZnO (ZnMgO) via in-situ reductive treatment of water-vapor with Al electrode, which can be used as electron-injection layer (EIL) for QLED or other optoelectronic devices. The resulting QLEDs exhibit optimal efficiency and extraordinarily-high brightness. This breakthrough work not only effectively eliminates the long-standing issue of device positive aging induced by acid-containing encapsulation, but also makes QLED lighting possible. I recommend it be published after minor revisions. Some issues should be addressed:

1. It is currently widely believed that there is excessive electron injection and insufficient hole injection in devices, and many studies have shown that improving hole injection or suppressing electron injection can enhance device performance (Nature, 2014, 515, 96–99; Sci. Adv. 10, 2024, eado0614; Adv. Mater. 2023, 2303950). However, in studies on positive aging, the general conclusion is that positive aging originates from the increased conductivity of the electron transport layer (ETL), which enhances electron transport efficiency. This study focuses on improving the conductivity of ZnMgO, and the resulting devices exhibit excellent performance (optimal efficiency and extraordinarily-high brightness), and successfully eliminates the positive aging issue. How could this seemingly contradictory phenomenon be explained?
2. ZnO has stronger n-type characteristics and better electron transport properties than ZnMgO. Why did the authors choose not to use ZnO in this work? What was the rationale behind this choice?
3. EPR is a powerful tool for analyzing the magnetic properties, defects, and free charge carriers in materials by detecting the spin characteristics of unpaired electrons. However, it is difficult to distinguish by EPR signal whether these unpaired electrons originate from defects / impurities, or from conduction band states. It is suggested to complement this with temperature-dependent conductivity or Hall effect experiments to investigate n-type characteristics of ZnMgO or ZnO films. In this way, the conductivity or mobility of ZnMgO or ZnO films treated under different conditions should be directly given.
4. After water-vapor treatment of ZnMgO/Al, in what form does the Al electrode exist after losing electrons? Does it exist as AlO or Al₂O₃? Will this affect the energy

levels and conductivity of ZnMgO? For QLED devices, the energy level configurations of the electrode, HIL and quantum dot light-emitting layer is another key factor regulating electron injection/transport. It is recommended to provide the energy level arrangement of QD, ZnMgO before and after treatment and Al electrode (including Ti, Au and Ag) and add the corresponding discussion.

5. In Figure 2g, the current density increases sharply above ~ 0.6 V but decreases below this voltage during the initial water-vapor treatment. The authors attribute that the in situ reductive treatment with the Al electrode greatly improves both hole blockage and electron conductivity of the ZnMgO EIL, which are respectively associated with deep recombination centers and free-electron concentrations within the ZnMgO EIL. However, for QLED devices, in the low voltage region (~ 0.7 eV) the current is mainly contributed by electrons and hole filling, and the holes have not yet been injected into the QD layer. Therefore, there seems to be no effect on hole transport at this time. This issue should be addressed further.
6. For QLED, the ultimate factor affecting its performance is the injection of electrons and holes into the light-emitting layer of quantum dots. The authors significantly improved the electrical conductivity of ZnMgO HIL through a simple in-situ reductive treatment, resulting in unprecedented device performance. In order to be able to apply this in-situ reductive doping method widely, the mechanism of n-type doping and the resulting efficient injection of charge into the quantum dot layer should be further discussed in detail.
7. This work has demonstrated ultra-bright and energy-efficient QLEDs across the visible spectrum. In Figure 4, all of the three red, green and blue QLED devices show ultra-bright and efficient at very high driving voltages. What is the possible mechanism?